# Glycolytic flux control by drugging phosphoglycolate phosphatase

Elisabeth Jeanclos [1,9], Jan Schlötzer[2,9], Kerstin Hadamek[1], Natalia Yuan-Chen [3], Mohammad Alwahsh[4,5,6], Robert Hollmann[4], Stefanie Fratz[1], Dilan Yesilyurt-Gerhards[1], Tina Frankenbach[1], Daria Engelmann[1], Angelika Keller[1], Alexandra Kaestner[1], Werner Schmitz [7], Martin Neuenschwander [8], Roland Hergenröder[4], Christoph Sotriffer [3], Jens Peter von Kries[8], Hermann Schindelin [2] & Antje Gohla [1] ✉

Targeting the intrinsic metabolism of immune or tumor cells is a therapeutic strategy in autoimmunity, chronic inflammation or cancer. Metabolite repair enzymes may represent an alternative target class for selective metabolic inhibition, but pharmacological tools to test this concept are needed. Here, we demonstrate that phosphoglycolate phosphatase (PGP), a prototypical metabolite repair enzyme in glycolysis, is a pharmacologically actionable target. Using a combination of small molecule screening, protein crystallography, molecular dynamics simulations and NMR metabolomics, we discover and analyze a compound (CP1) that inhibits PGP with high selectivity and sub-micromolar potency. CP1 locks the phosphatase in a catalytically inactive conformation, dampens glycolytic flux, and phenocopies effects of cellular PGP-deficiency. This study provides key insights into effective and precise PGP targeting, at the same time validating an allosteric approach to control glycolysis that could advance discoveries of innovative therapeutic candidates.

Inhibiting glycolysis is an important strategy to modulate immune responses and to treat cancer, but therapeutic progress has been limited so far[1–3]. It is an open question whether all metabolic targets of interest have been exhausted[3]. Recently, a previously unrecognized layer of control has emerged with the identification of metabolite repair enzymes that maintain proper flux in central carbon metabolism[4,5], but pharmacological approaches to target these enzymes are still lacking.

Metabolic enzymes, including those involved in glycolysis, have long been regarded as perfect catalysts, but it is now clear that they can process non-cognate substrates, catalyze different types of reactions, or mediate side-reactions that produce accidental metabolites[6,7]. Such metabolic by-products can be wasteful or even toxic, either because they are inherently reactive, or because they resemble main substrates or effectors of other enzymes and cause their inhibition. Numerous enzymes have now been identified that specifically repair problematic metabolites and thus mitigate metabolite damage[4,5]. In glycolysis alone, 10 such metabolite repair proteins work in conjunction with 10 glycolytic core enzymes to maintain proper pathway flux, and inborn errors in some of these repair enzymes can cause severe human diseases[8,9]. Specific chemical tools to target repair enzymes in glucose metabolism are largely lacking, but could facilitate further studies on their roles in physiology and disease.

Phosphoglycolate phosphatase (PGP, also called glycerol 3-phosphate phosphatase[10] or AUM[11]) is an evolutionarily conserved

[1]Institute of Pharmacology and Toxicology, University of Würzburg, Würzburg, Germany. [2]Rudolf Virchow Center for Integrative and Translational Bioimaging, University of Würzburg, Würzburg, Germany. [3]Institute of Pharmacy and Food Chemistry, University of Würzburg, Würzburg, Germany. [4]Leibniz-Institut für analytische Wissenschaften-ISAS, Dortmund, Germany. [5]Institute of Pathology and Medical Research Center (ZMF), University Medical Center Mannheim, Heidelberg University, Mannheim, Germany. [6]Department of Pharmacy, Faculty of Pharmacy, Al-Zaytoonah University of Jordan, Amman, Jordan. [7]Department of Biochemistry and Molecular Biology, Theodor Boveri Institute, Biocenter, University of Würzburg, Würzburg, Germany. [8]Leibniz-Forschungsinstitut für Molekulare Pharmakologie-FMP, Berlin, Germany. [9]These authors contributed equally: Elisabeth Jeanclos, Jan Schlötzer. ✉e-mail: antje.gohla@uni-wuerzburg.de

metabolite damage control enzyme that can dephosphorylate and thereby repair three adventitious glycolytic metabolites: 2-phospho-L-lactate, 4-phosphoerythronate[12], and 2-phosphoglycolate[12–14]. An accumulation of 2-phospho-L-lactate or of 4-phosphoerythronate inhibits glycolysis or pentose phosphate pathway flux, respectively[12]. PGP is the only mammalian enzyme that has been described to act on 2-phosphoglycolate, 2-phospho-L-lactate and 4-phosphoerythronate, and there are no known functional overlaps with other metabolite repair processes in the context of glycolysis[8,12] (see also Supplementary Note 1). The whole-body genetic inactivation of *Pgp* in mice caused lethal growth defects in midgestation and blocked the proliferation of embryonic fibroblasts, whereas the targeted PGP inactivation in e.g. endothelial cells during embryogenesis and adulthood did not lead to an overt phenotype[13]. These observations are consistent with a fundamental role of PGP in balancing glycolytic and pentose phosphate pathway flux in rapidly proliferating cells, thus suggesting that PGP could be an interesting target to probe the physiological and pathological roles of a glycolysis-associated metabolite repair protein.

PGP belongs to the haloacid dehalogenase (HAD)-type superfamily of phosphatases, a large, widespread and evolutionarily ancient class of hydrolases with emerging roles in human physiology and disease[15–18]. PGP consists of a catalytic core harboring a Rossmanoid fold, and a large cap domain restricting access to the active site[11]. Commonly used phosphatase inhibitors do not inhibit HAD phosphatases, and natural inhibitors that may serve as lead compounds are unknown. Furthermore, chemical tools to specifically target mammalian HAD phosphatases are scarce[19–21], and small molecules that selectively inhibit PGP have not yet been described.

Here, we report the identification and extensive biochemical, structural and functional evaluation of CP1, a specific small-molecule PGP inhibitor that blocks PGP-dependent glycolytic flux in mammalian cells. Our study thus validates a pharmacological target class and lays the groundwork for the development of innovative therapeutic approaches to target cellular metabolism.

## Results

### A high-throughput screening campaign identifies PGP inhibitors

We screened the FMP small molecule repository containing 41,536 compounds for molecules able to modulate the phosphatase activity of recombinant, highly purified murine PGP (Supplementary Fig. 1a; see Methods for further details). Difluoro-4-methylumbelliferyl phosphate (DiFMUP) was used as a fluorogenic phosphatase substrate in a primary screen, and a counterscreen was performed with pyridoxal 5'-phosphate phosphatase (PDXP), the closest PGP relative[11]. The PGP inhibitor hits that were inactive against PDXP were subsequently validated in a secondary assay, using 2-phosphoglycolate (PG) as a physiological PGP substrate. $IC_{50}$ values were determined for the 23 PGP inhibitor hits that blocked PG dephosphorylation by >50% (Supplementary Table 1), and three of the most potent PGP inhibitors (compounds 1-3/CP1-3) were selected for further characterization.

As shown in Fig. 1a, CP1-3 potently inhibited the activity of both human and murine PGP towards PG. CP1-3 also inhibited the activity of murine PGP towards the generic phosphatase substrate para-nitrophenylphosphate (pNPP). Furthermore, CP1 effectively inhibited the dephosphorylation of 2-phospho-L-lactate (2PL), 4-phosphoerythronate (4PE) and glycerol 3-phosphate (G3P; see also Supplementary Fig. 1b). We confirmed that CP1-3 did not inhibit PDXP activity towards pyridoxal 5'-phosphate (PLP)[22]. CP1-3 were also able to block PG dephosphorylation catalyzed by an artificial protein consisting of the PDXP catalytic core and the PGP specificity-determining cap (PDXP/PGP hybrid[11]). However, for CP1, ~37-fold higher concentrations were required for a half-maximal inhibition of the murine PDXP/PGP hybrid compared to full-length murine PGP. These data suggest that CP1-3 interact with the PGP cap domain, and

that binding to both the cap and the core domain is essential for an optimal inhibition of PGP by CP1.

To exclude the possibility that the inhibitory effects of CP1-3 were caused by protein denaturation, we investigated the thermal stability of PGP in the absence or presence of increasing compound concentrations by differential scanning fluorimetry. Supplementary Fig. 1c shows that the inhibitors in fact stabilized PGP in a concentration-dependent manner, as reflected by an increase in the maximal PGP melting temperature from 66.4 °C (control conditions) by 4.6°, 2.2°, or 7.1° for CP1, −2, or −3, respectively. Thus, the observed PGP inhibition was not due to protein destabilization.

We performed isothermal titration calorimetry (ITC) experiments to further characterize inhibitor binding to PGP. CP1 and CP3 showed a relatively high-affinity interaction with PGP characterized by $K_D$-values of 0.78 and 0.43 μM, respectively, and bound PGP with monovalent binding stoichiometries (Fig. 1b). CP2 binding to PGP could not be analyzed by ITC due to its limited solubility.

### Structure-activity relationships

CP1 and CP2 are structurally related diphenylether and diphenylmethylene dibenzoic acid derivatives with similar inhibitory properties (Fig. 1a). Taking these hits as model compounds, we examined whether commercially available analogs might be more potent PGP inhibitors. We tested a pyrrolidinedione-derivative (1B), two isomers (1C and 1D) and two isomeric fragments (1E and 1F) of CP1, as well as four CP2 analogs (Supplementary Fig. 2a) in the PG dephosphorylation assay. However, none of the tested CP1 analogs were anywhere as potent as CP1 (Supplementary Fig. 2b). In contrast, the CP2 analogs with the exception of 2E still retained inhibitory potency, with 2B being even slightly more potent than the parental compound (Supplementary Fig. 2b).

### Selectivity of PGP inhibitors

PGP is a member of the vast and ancient class of haloacid dehalogenase (HAD)-type phosphatases. The overall structure of HAD phosphatases and their catalytic residues are highly conserved. All HAD phosphatases consist of a catalytic core domain with a Rossmanoid-fold. This core is accessorized with so-called cap domains, which are mobile elements that can shield the entry to the catalytic cleft. The cap size (tiny C0 caps versus larger C1 or C2 type caps), their structure and insertion site determine the substrate binding pocket of the respective enzyme: C0-type HAD phosphatases can dephosphorylate macromolecules, whereas C1/C2-type HAD phosphatases are typically directed against small molecules that are able to enter the catalytic cleft. In addition, the caps contain substrate specificity loops that interact with the substrates. Hence, cap types are key differentiators of HAD phosphatases[15,16,23,24].

To explore the selectivity of CP1-3 for PGP, a C2-capped phosphatase, we tested 12 other mammalian HAD phosphatases, representing members of all three cap subtypes (Fig. 1c). In addition, we analyzed the activity of CP1-3 towards the serine/threonine phosphatase calcineurin (PP2B), the prototypical tyrosine phosphatase PTP1B, and the broadly acting calf intestinal alkaline phosphatase (CIP). When assayed at a concentration of 40 μM (i.e., ~25- to 110-fold above the respective $IC_{50}$ values for PG-dephosphorylation by murine PGP, see Fig. 1a), CP1-3 were inactive against nine of the tested HAD phosphatases. In addition to PGP, CP2 inhibited the C1-capped N-acetylneuraminate 9-phosphate phosphatase NANP/HDHD4 ($IC_{50}$ ~1 μM; Supplementary Fig. 2d). CP1 and −2 inhibited the C1-capped soluble cytosolic 5'-nucleotidase 1A (NT5C1A), albeit with $IC_{50}$ values ~57- or 14-fold above those for PGP, respectively (Fig. 1c, Supplementary Fig. 2d). In contrast, the NT5C1A-related 5'-(3')-deoxyribonucleotidase/inositol 5'-monophosphate (IMP) phosphatase NT5C and the mitochondrial 5'-(3')-deoxyribonucleotidase/uridine 5'-monophosphate (UMP) phosphatase NT5M were not inhibited by CP1 or −2. Compound 3 very

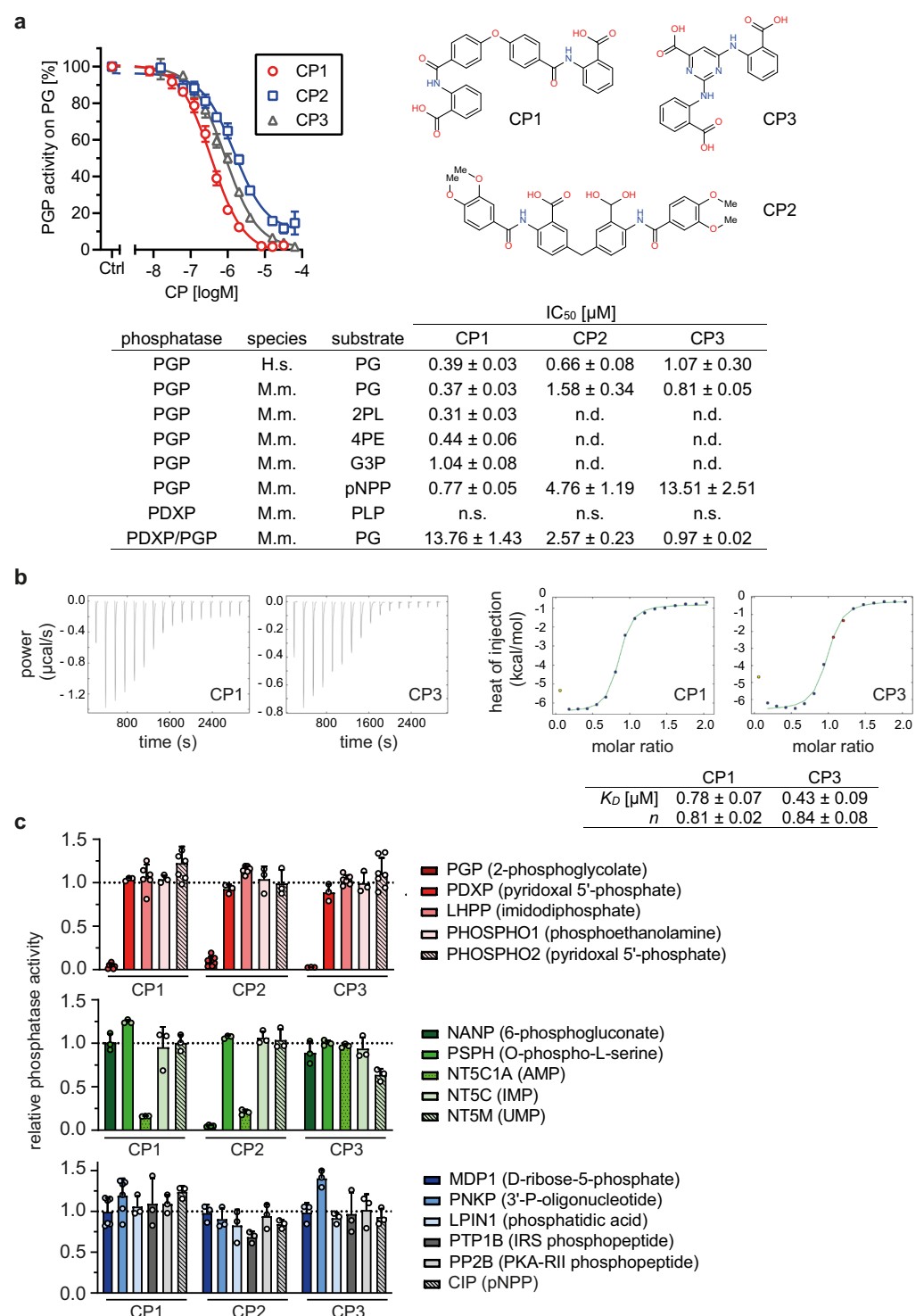

**Fig. 1 | Identification and characterization of phosphoglycolate phosphatase (PGP) inhibitors. a** Determination of half-maximal inhibitory constants (IC$_{50}$) of compounds **1-3** (CP1-3, see 2D-structures on the right), using purified murine PGP and phosphoglycolate (PG) as a substrate. Error bars not shown are hidden by the symbols. The table gives IC$_{50}$ values (in μM) of CP1-3 for purified PGP, PDXP, or the PDXP/PGP hybrid protein composed of the PDXP catalytic core and the PGP cap domain. H.s., Homo sapiens; M.m., Mus musculus, n.d., not determined, n.s., no significant inhibition detectable. All data are mean values ± S.E.M of at least $n$ = 3 biologically independent experiments. Inhibition of human PGP-catalyzed PG dephosphorylation by CP2 or CP3 was assessed in $n$ = 5 or $n$ = 4 independent experiments, respectively. Inhibition of murine PGP-catalyzed PG dephosphorylation by CP2 and CP3, and of murine PGP-catalyzed G3P dephosphorylation by CP1 was determined in $n$ = 4 independent experiments. **b** Isothermal titration calorimetry measurements of the association reactions of CP1 or CP3 with murine PGP. The left panels show representative thermograms, the right panels show fitted binding curves of the isotherms. The binding data to PGP (mean values ± S.E.M from $n$ = 3 independent experiments) are listed in the table. $K_D$, dissociation constant; $n$, stoichiometry. **c** In vitro activity assays with the indicated phosphatases in the presence of 40 μM CP1, CP2 or CP3. Upper panel, C2-capped HAD phosphatases; middle panel: C1-capped HAD phosphatases; lower panel: C0-capped HAD- and other phosphatases. The respective phosphatase substrates are given in the legend. Phosphatase activities in the presence of CP1-3 were normalized to the respective enzyme activities measured in the presence of the DMSO solvent control. Data are mean values ± S.E.M.; the number of $n$ biologically independent experiments is indicated by the number of open symbols. Source data are available as Source Data File.

## Table 1 | Data collection and refinement statistics

| | PGP$^{D34N, C297S}$ | PGP$^{D34N, C297S}$ + CP1 |
|---|---|---|
| Synchrotron | ESRF | PETRA III |
| Beamline | ID23-1 | P13 |
| Wavelength (Å) | 0.9724 | 0.9763 |
| Space group | P1 | P4$_1$ |
| Unit cell parameters (a, b, c (Å); α, β, γ (°)) | 84.26 91.20 101.32 103.46 89.98 104.63 | 105.9 105.9 83.3 90.0 90.0 90.0 |
| Resolution limits (Å) [a] | 49.17-2.31 (2.54-2.31) | 47.36-3.16 (3.38-3.16) |
| R$_{merge}$ [b] | 0.094 (1.078) | 0.229 (2.334) |
| R$_{pim}$ [c] | 0.062 (0.684) | 0.087 (0.869) |
| CC$_{1/2}$ | 0.993 (0.281) | 0.993 (0.302) |
| Multiplicity | 3.1 (3.4) | 7.8 (8.0) |
| Unique Reflections | 88,859 (4,443) | 13,010 (651) |
| Completeness elliptical / spherical (%) | 91.42 (59.14) / 71.32 (14.39) | 90.85 (46.06) / 82.19 (23.33) |
| <I/σ > [d] | 7.43 (1.23) | 7.59 (1.10) |
| R$_{work}$/R$_{free}$ [e,f] | 0.2068/0.2308 | 0.1984/0.2420 |
| Deviations from ideal values in: | | |
| Bond distances (Å) | 0.005 | 0.003 |
| Bond angles (°) | 0.846 | 0.735 |
| Planarity (Å) | 0.006 | 0.005 |
| Dihedral angles (°) | 16.53 | 19.03 |
| Ramachandran statistics (%) [g] | 98.41/1.47/0.12 | 98.37/1.63/0.0 |
| Average B-factor (Å$^2$) | 71.3 | 89.5 |

[a] Numbers in parentheses refer to the respective highest resolution data shell in each data set.

[b] $R_{merge} = \Sigma_{hkl}\Sigma_i \mid I_{hkl,i} - <I_{hkl}> \mid / \Sigma_{hkl}\Sigma_i I_{hkl,i}$ where $<I_{hkl}>$ represent the average of the i observations of reflection $hkl$.

[c] $R_{p.i.m.} = \Sigma_{hkl}(1/(n-1))^{-1/2}\Sigma_i \mid I_{hkl,i} - <I_{hkl}> \mid / \Sigma_{hkl}\Sigma_i I_{hkl,i}$ where $n$ is the multiplicity of the observed reflection.

[d] Indicates the average of the intensity divided by its S.D. value.

[e] $R_{work} = \Sigma \mid F_o - F_c \mid / \Sigma \mid F_o \mid$ where $F_o$ and $F_c$ are the observed and calculated structure factor amplitudes.

[f] $R_{free}$ same as $R_{work}$ for 2.5% (apo) and 5% (inhibitor) of the data randomly omitted from refinement.

[g] Ramachandran statistics indicate the fraction of residues in the favored, allowed, and disallowed regions of the Ramachandran diagram as defined by MolProbity[71].

weakly inhibited NT5M. None of the compounds inhibited PP2B, PTP1B or CIP. Together, these results indicate that CP1-3 are remarkably selective for PGP.

### Mode of PGP inhibition

To probe the mechanism of PGP inhibition, we assayed the steady state kinetics of PG dephosphorylation in the presence of increasing CP1-3 concentrations (Supplementary Table 2). Analysis of the derived kinetic constants suggested different modes of PGP inhibition. CP1 increased the $K_M$ up to ~3-fold, and reduced $v_{max}$ values ~3-fold, whereas CP2 and −3 increased the $K_M$ between ~3-fold without affecting $v_{max}$. Thus, CP1 shows a mixed mode of inhibition (apparently between competitive and non-competitive), and CP2 and −3 act as competitive inhibitors.

### Observation of open and closed active site conformations in PGP

To understand mechanisms of PGP inhibition on a molecular level, we set out to obtain structural information of PGP in complex with an inhibitor. As a first step, we defined suitable crystallization conditions for full-length PGP, whose structure was previously unknown. We had shown earlier that PGP exists in a dynamic equilibrium between homodimers and -tetramers[11], and that the mutation of Cys297 to Ser (PGP$^{C297S}$) abolished disulfide-mediated PGP tetramer formation[14]. We therefore hypothesized that PGP$^{C297S}$ might be a suitable PGP variant for crystallization. We verified that PGP$^{C297S}$ was enzymatically active

and sensitive to compound 1 (PGP$^{C297S}$ IC$_{50}$ ~ 0.25 ± 0.05 μM; Supplementary Fig. 3a, b), and that CP1 did not affect PGP$^{C297S}$ self-association (Supplementary Fig. 3c; see below). For subsequent crystallization attempts, we additionally exchanged the catalytically essential Asp34 to Asn, which abolishes PGP activity towards its substrates[11].

Full-length murine PGP$^{D34N/C297S}$ (here referred to as apo-PGP) crystallized in space group P1 with four dimers per asymmetric unit (Supplementary Fig. 4), referred to as AB, CD, EF and GH dimers. The structure was solved by molecular replacement with the murine PDXP/PGP hybrid protein[11] (PDB code 4BKM). The structure was rebuilt and refined against data truncated anisotropically with Staraniso (http://staraniso.globalphasing.org/cgi-bin/staraniso.cgi) to 2.31/2.74 Å in the best/worst direction, resulting in an $R_{work}$ of 0.2068 and an $R_{free}$ of 0.2308 (PDB code 7PO7). Data collection and refinement statistics are summarized in Table 1. While monomers A-F and H are well defined in the electron density maps, large sections of the catalytic core of monomer G display very weak density. The electron density map of the active site of monomer E is shown in Supplementary Fig. 5.

Figure 2a shows the structure of the EF-dimer of full-length apo-PGP. Like its paralog PDXP[22] and the PDXP core/PGP cap hybrid protein[11], PGP homodimerizes via its cap domain. Aligning the core domains (Cα atoms of residues 11–99 & 238–320) of the B-F and H monomers (the G-monomer was omitted from the analysis due to its poor density for the core domain) on the A-monomer resulted in a close superposition with values for the root mean square deviation (RMSD) between 0.41 Å and 0.50 Å for the Cα atoms for monomers B, C, F and H and slightly higher deviations of 0.58 Å and 0.62 Å for monomers D and E. The disparity in the alignment for monomers D and E correlates with the presence of a distinct conformation of the cap domain in both monomers. An analysis with the DynDom server[25] revealed that in five (chains A-C and F-H) of the seven PGP monomers which were part of the analysis, the cap domain was rotated by ~11–15° relative to the conformation present in the other two monomers (chains D and E). Coupled to this rotation is a shift of the substrate specificity loop (amino acids 202–215) by up to ~6 Å for corresponding Cα atoms towards the catalytic core. Conversely, the cap domains (Cα atoms of residues 100–237) can be superimposed, which then identifies a movement of the PGP 'flap' (amino acids 39–46 of the core domain) by >5 Å towards the catalytic core (Fig. 2b, c). The conformational changes do not appear to be induced by crystal packing forces since the cap domains of monomers D and E as well as that of monomer A, which together represent both conformations, are hardly involved in crystal packing contacts in contrast to the more extensively participating monomers B, C, F and H (Supplementary Fig. 4).

Flap and cap movements are known to gate access to the catalytic site in C1-capped HAD phosphatases[26]. An inspection of the PGP catalytic cleft showed that the active site of PGP is organized as a channel, and that Asn34 (corresponding to the Asp34 nucleophile) is centrally buried within this cleft (Fig. 2d). Upon rotation of the substrate specificity loop and the flap, channel entrances become obstructed, and access to the active site is blocked. To our knowledge, experimental proof for the existence of open (chains D and E) and closed (chains A-C, F and H) conformations in a C2-capped mammalian phosphatase has not been previously provided.

### Co-crystal structure of PGP bound to CP1

Because of its relatively high potency, selectivity profile, and high-affinity binding of CP1 to PGP (Fig. 1), we focused our structural analyses on this inhibitor. CP1-bound PGP crystallized in the tetragonal space group P4$_1$ with one dimer per asymmetric unit. The structure was solved by molecular replacement with the AB dimer of the apo-PGP structure, followed by refinement against anisotropically truncated data (again with Staraniso) to resolutions of 3.16/3.41 Å in the

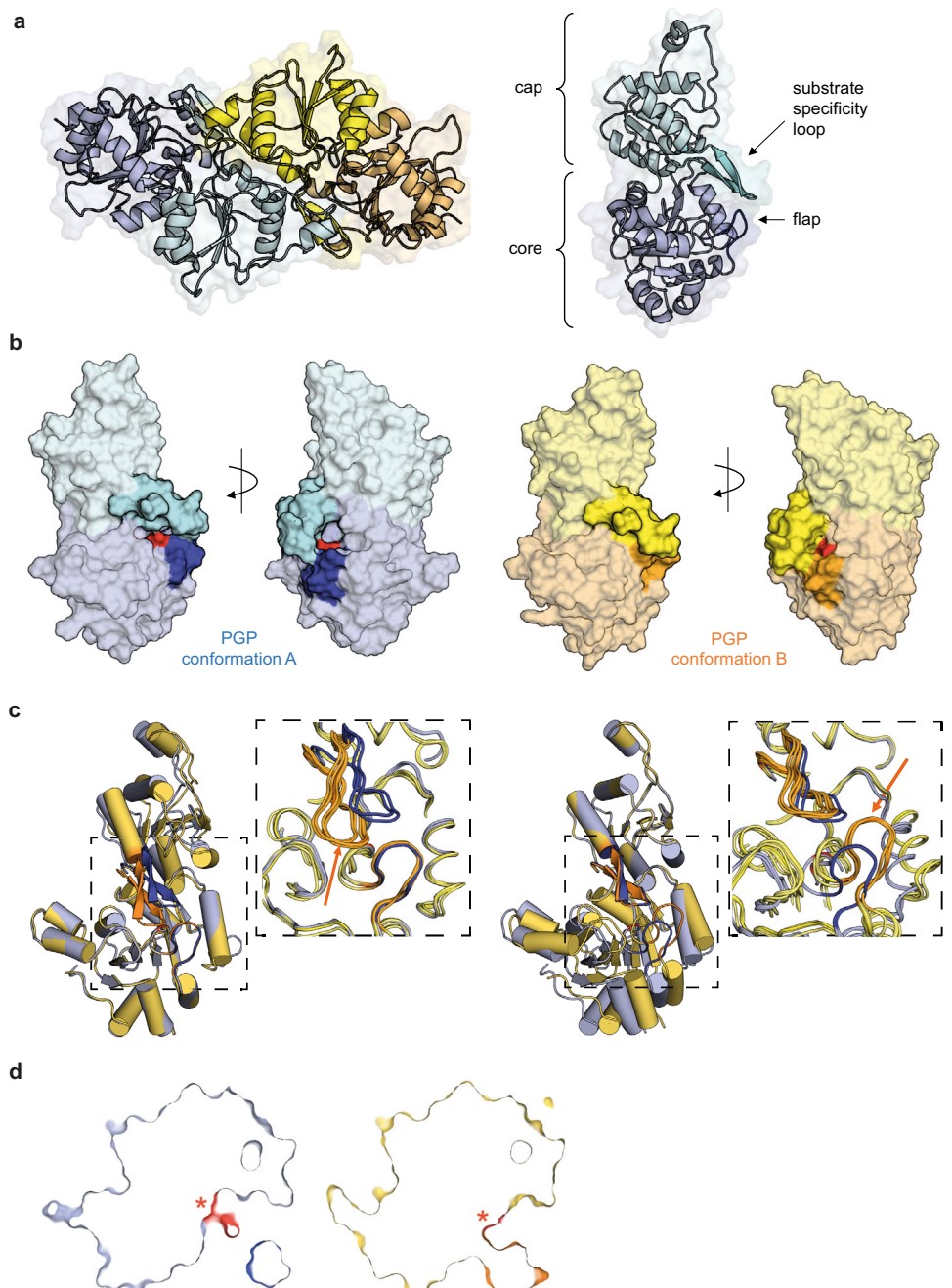

**Fig. 2 | Observation of open and closed active site conformations in apo-PGP.**
**a** Left, X-ray crystal structure of homodimeric, full-length murine PGP[D34N/C297S] (apo-PGP; chain E in blue-green, chain F in yellow-gold), refined to a resolution of 2.3 Å (PDB code 7PO7). Right, ribbon representation of chain E; important structural features are indicated. **b** Alignment of the catalytic core domains of apo-PGP monomers. Left, surface representation of chain E (labeled in blue-light blue). Right, surface representation of chain A (labeled in yellow-gold). The substrate specificity loops are marked in turquoise and bright yellow, the flap domains are labeled in dark blue and orange, and the catalytic aspartate (Asn34 in apo-PGP) is shown in red. In chain A the cap domain is rotated relative to chain E, resulting in a shift of the substrate specificity loop towards the catalytic core. The views differ by a 180° rotation as indicated. **c** Superposition of apo-PGP monomers (chain A in

yellow, chain E in blue) after aligning the core domains (left panel) or the cap domains (right panel). The boxed areas are enlarged and include a superposition of all monomers of the asymmetric unit. The movement of the substrate specificity loop (left panel) or the flap domain (right panel) in chain A towards the catalytic core is highlighted with arrows. **d** Catalytic cleft architectures of chain E (blue) and chain A (yellow). Asn34 (marked in red and highlighted with a red asterisk; corresponding to the Asp34 nucleophile) is centrally buried within the active site channel. Shown are slices through surface representations of the catalytic cleft. The dark blue color in chain E and the orange color in chain A correspond to the substrate specificity loop/flap domains. The rotation of the substrate specificity loop and the flap, which is observed in chains A-C, F and H, leads to an obstruction of the active site channel entrance and blocks access to the active site.

best (a*b*-plane) and worst direction (along c*), resulting in an $R_{work}$ of 0.1984 and an $R_{free}$ of 0.2420 (Fig. 3a and Table 1; PDB code 7POE). Already after molecular replacement both protomers showed clear density for CP1 (Fig. 3b). The electron density map of the active site of

CP1-bound monomer A is shown in Supplementary Fig. 6. The temperature factors of the atoms in CP1 were similar in magnitude to those of the surrounding atoms, therefore suggesting that the inhibitor is bound with an estimated occupancy of ~1.

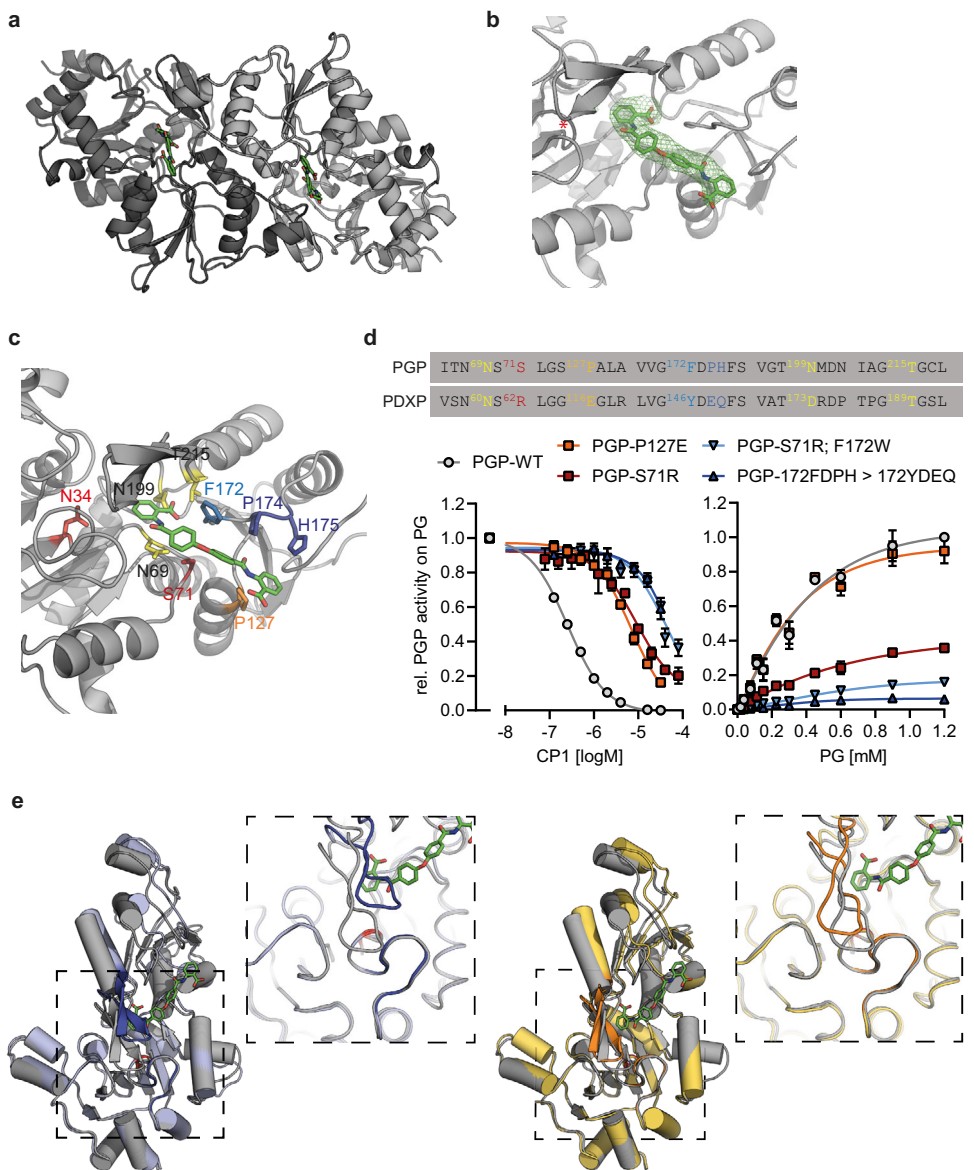

**Fig. 3 | Co-crystal structure of PGP in complex with CP1. a** X-ray crystal structure of homodimeric, full-length murine PGP$^{D34N/C297S}$ bound to CP1 (PDB code 7POE). The two protomers are shown in light or dark grey; CP1 is shown in green. **b** The $F_o − F_c$ polder electron density map of the inhibitor is contoured at an RMSD of 3.0 in green mesh, and superimposed with the refined model. The position of the catalytic Asp/Asn34 is highlighted with a red star. **c** Detailed structure of bound CP1 and adjacent residues of the active site. **d** Verification of CP1 – PGP interactions. The PGP residues highlighted in the alignment were exchanged for the corresponding amino acids of the CP1-insensitive PGP paralog PDXP. Left panel, determination of the IC$_{50}$ values of CP1 for purified PGP$^{WT}$ or the indicated PGP variants. Right panel,

relative phosphatase activity of purified PGP$^{WT}$ or the indicated PGP variants at different PG concentrations. All data are mean values ± S.E.M of $n = 3$ independent experiments. Error bars not shown are hidden by the symbols. **e** Alignment of the CP1-bound PGP structure (grey) with apo-PGP conformers in the open (blue, chain E) or closed (yellow, chain A) active site conformation. The catalytic Asp/Asn34 is highlighted in red (visible in the open conformation); inserts show a magnification of the marked area showing the substrate specificity loops and flap domains. The CP1-bound structure adopts a cap-closed conformation, with the PGP substrate binding loop rotated towards the active site and additionally moved outwards. Source data are available as Source Data File.

CP1 binds adjacent to, but not directly into, the PGP active site with one of the benzoic acid moieties (referred to in the following as proximal) at a distance of just below 6 Å from the catalytic Asp34 (Asn34). Inhibitor binding appears to be primarily stabilized by non-polar interactions and a very small number of polar interactions (Fig. 3c). The proximal benzoic acid moiety of CP1 forms the only direct hydrogen bonds via its carboxylic acid to the nitrogen atom of the carboxamide function of Asn199 and to the side chain hydroxyl group of Thr215. At the limited resolution of this structure these contacts should be considered potential interactions. Overall, CP1 is embedded in a cavity that extends from the active site of monomer A towards the dimerization interface with monomer B. One side of this

cavity is formed by the more polar residues 35–37, 68–71, 125–127 and 143–145, which originate entirely from monomer A. In contrast, the opposite side is formed by both monomers and is generally more hydrophobic in character. Key residues include stretches 172–175, 201–204/212–215 within monomer A, alongside residues 151–156 of monomer B. The primary driving force for binding seem to be van der Waals interactions involving the side chains of the non-polar residues Pro127, Phe172, Pro174, Leu204 and Trp156 of monomer A. In addition, the non-polar atoms of Asp36, Asn69, Ser71 and His175 engage in van der Waals interactions with CP1. All PGP residues found to engage in CP1 interactions (see also Fig. 4) are identical in murine and human PGP (Supplementary Fig. 7).

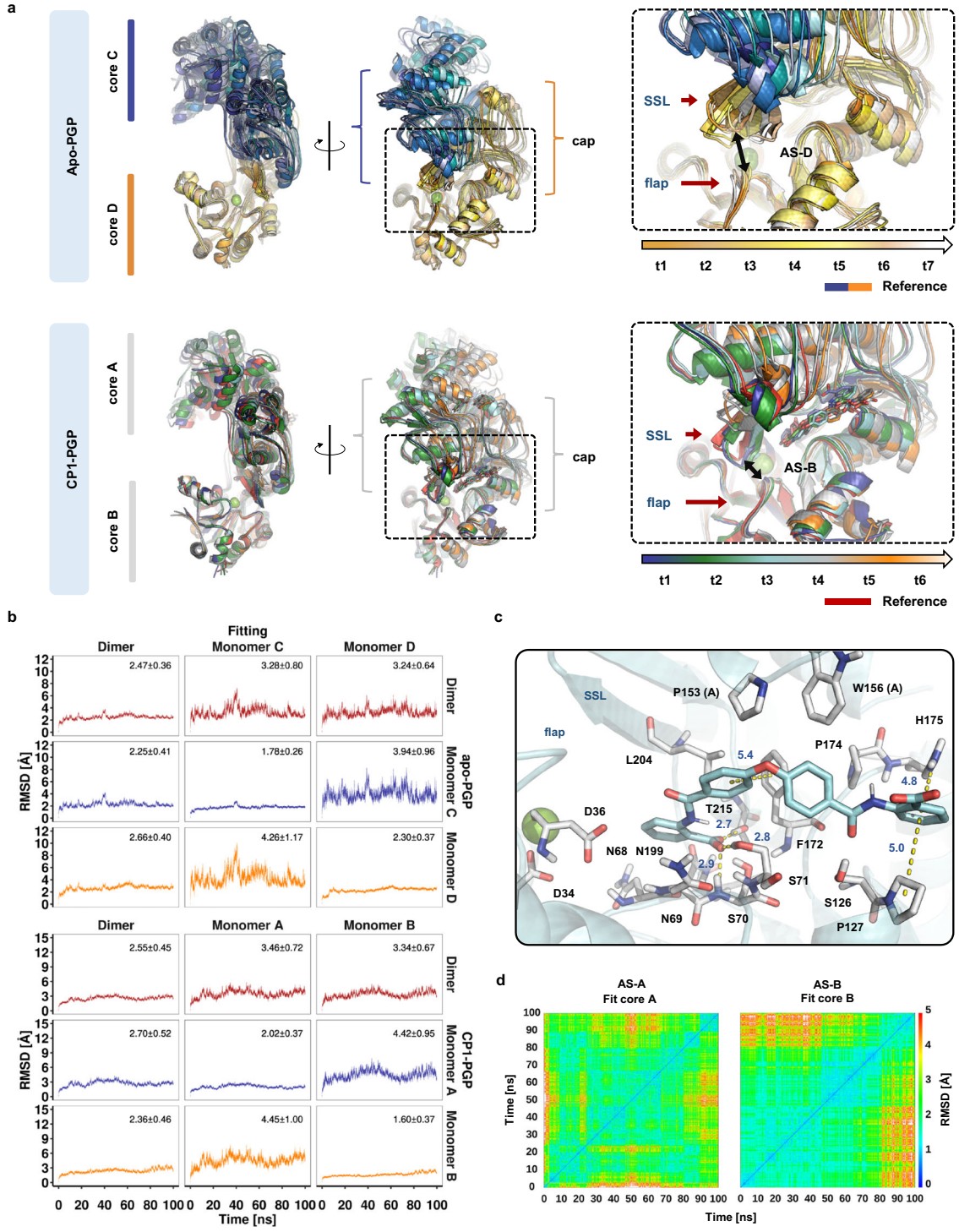

**Fig. 4 | CP1 maintains PGP in a cap-closed state during MD simulation.**
**a** Superposition of apo-PGP (upper panel) and CP1-PGP (lower panel) conformations along the trajectory, using core residues of monomer D (apo-PGP) or monomer B (CP1-PGP) for alignment. Snapshots were taken from specific time intervals in which the protein adopted distinct conformations. The sampled conformations from different time points ($t_1$ to $t_{6/7}$) are colored based on a progressing timeline illustrated by arrows below the right panels. The global view in the left panel shows how the cores are stable throughout the simulation, whereas their caps and counter-chains fluctuate. The magnification of the binding site highlights a closed catalytic site access gate, controlled by the substrate specificity loop (SSL) and flap domains in CP1-PGP. In contrast, the SSL has subtly shifted away from the flap domain in the apo-PGP structure, resulting in a broader active site. **b** RMSD timelines of the backbone atoms with respect to the starting structure, along with the corresponding average and standard deviation. The RMSD values were

calculated for different parts of the homodimer, as indicated by the labels at the right side of the diagrams: the entire dimer (top row) or each of the two monomers (middle and bottom row). Different fittings were used for the calculations, as indicated by the labels above the diagrams: Left column, fitting of the trajectory onto the dimer; middle and right column: fittings on the individual monomers. The comparison reveals high stability and low fluctuations within each monomer (in particular within the core domain), but a dynamic dimer ensemble with motions of the two monomers relative to each other. **c** Binding mode with highlighted interactions of CP1 at the active site of monomer B from snapshot t3 (cf. A). Distances are given in Å and depicted by yellow dashes. The $Mg^{2+}$ ion is shown as a green sphere. **d** 2D-RMSD plots for CP1 at the active sites of monomer A (left plot) and monomer B (right plot). The RMSD was calculated for the CP1 heavy atoms after fitting the trajectory onto the core-domain backbone atoms.

Towards the end of the refinement, residual density could be tentatively interpreted with a second CP1 molecule, which is binding to each monomer on the opposite side of PGP. The binding site is quite remote from the active site (closest distance of ~15 Å) and is in close proximity to the non-crystallographic twofold axis of symmetry. While the ITC data indicate binding of a single copy of CP1 to each monomer (Fig. 1), it cannot be ruled out that this second binding event does not lead to a change in enthalpy. The crystallographic evidence documents, however, that the second CP1 molecule is weakly bound and hence no biological significance was attributed to the second binding site.

To verify the putative CP1 – PGP interactions in the primary binding site, we introduced single or multiple mutations into the binding interface by exchanging PGP residues for the corresponding amino acids of the CP1-insensitive PGP paralog PDXP (Fig. 3d). Hence, Ser71 was changed to the large, positively charged Arg (PGP$^{S71R}$), Pro127 was altered to Glu (PGP$^{P127E}$), and the hydrophobic sequence $^{172}$FDPH was swapped with the respective residues in PDXP (PGP$^{172FDPH>YDEQ}$). In addition to the S71R variant, we also mutated Phe172 (facing Ser71 on the opposite side of the active cleft) to the less hydrophobic Trp (PGP$^{F172W}$). Figure 3d (left panel) shows that all mutations resulted in a strongly reduced potency of CP1, supporting the role of these residues in inhibitor binding. Exchanging both Ser71 and Phe172 (PGP$^{S71R; F172W}$) or ablating the hydrophobic pocket (PGP$^{172FDPH>YDEQ}$) increased the CP1 IC$_{50}$ value ~600-fold compared to PGP$^{WT}$. These data are consistent with the binding mode of CP1: a replacement of Ser71 with Arg obstructs the binding site, a Glu at position 127 as well as an Asp at position 174 (both instead of a Pro) are electrostatically repulsive with respect to the distal carboxylate of CP1, and a Trp instead of Phe172 is sterically too demanding for CP1 binding. Figure 3d (right panel) shows that also the phosphatase activity of PGP$^{S71R}$ was strongly decreased. This indicates a key role of Ser71 for active site accessibility. The almost complete loss of phosphatase activity in PGP-Phe172 mutants (PGP$^{S71R; F172W}$; PGP$^{172FDPH>YDEQ}$) might likewise be explained by impeded substrate entry into the active site channel due to side chain interference and charge repulsion effects. Interestingly, the phosphatase activity of PGP$^{P127E}$ was comparable to PGP$^{WT}$. Furthermore, ITC experiments demonstrated that the affinity of CP1 for PGP$^{P127E}$ was 15-fold lower than for PGP$^{WT}$ ($K_D$ 11.93 ± 0.41 μM; $n$ = 1.15 ± 0.04; mean values ± S.E.M. of two independent experiments; see Fig. 1b for comparison). This result is in line with the higher IC$_{50}$ of CP1 for PGP-P127E compared to PGP-WT (Fig. 3d, left panel). Hence, PGP$^{P127E}$ represents a PGP variant that retains full phosphatase activity while being CP1-resistant.

The comparison of the CP1-bound PGP structure with apo-PGP conformers with the DynDom server[25] revealed that CP1 – PGP adopts a conformation that resembles the cap-closed state (Fig. 3e). Domain movements were recognized for both subunits of the inhibitor-bound structure only with respect to chains D and E of the apo-structure, thus indicating that both monomers of the CP1-bound structure and the majority of apo-PGP monomers adopt the same conformational state. Despite being in the same conformational state, there are differences between the closed state in the apo and inhibitor bound structures, e.g. the substrate binding loop, rather than moving solely towards the active site, is also translated outwards. This additional movement is needed to create the space required for incorporating CP1 (Fig. 3e). Hence, CP1 appears to inhibit PGP activity by locking it in a cap-closed conformation.

Based on the inhibitor-bound structure and the competitive component of PGP inhibition by CP1 (increased $K_M$, see Supplementary Table 2), it seems likely that CP1 sterically hinders substrate access to the active site. In addition, the cap-closed conformation of CP1-bound PGP will block solvent entry into the enzymatic pocket, and thus prevent the hydrolysis of the phospho-aspartate intermediate during the final catalytic step. The resulting reduced rate of product formation is consistent with the observed non-competitive component of CP1-mediated PGP inhibition (reduction of $\nu_{max}$, see Supplementary Table 2).

## CP1 locks PGP in a closed-cap conformation during molecular dynamics simulations

To further investigate the PGP conformations as well as the interactions of CP1 at the active site, we carried out molecular dynamics (MD) simulations. For this purpose, PGP was restored to the wild-type active site with Asp34 and a Mg$^{2+}$ ion (see Methods). The crystal structures of the homodimeric apo-PGP (chains C and D) and CP1-bound PGP (chains A and B) were used as starting conformations to sample two 100 ns trajectories. The binding mode of CP1 to the wild-type active site was obtained by docking calculations, which confirmed the crystallographically obtained binding mode with an RMSD of ~0.5 Å (see Methods and Supplementary Figs. 8–11). Analyzing the protein RMSD profile along the trajectory based on different fittings using either the entire dimer or the individual monomers (Fig. 4b) revealed the overall stability of the monomers, but considerable fluctuations and conformational changes across the dimer interface, where the caps are located (see Supplementary Figs. 12–13 for further characterization of the dynamics by means of RMSF profiles and principal component analyses, and Supplementary Movies 1-4 related to Supplementary Fig. 13). Within these cap domains, the substrate specificity loop (SSL; responsible for the substrate access gate) was predominantly in a closed state in the CP1-PGP simulation, whereas it moved and sampled the open form in the simulation of the apo structure (Fig. 4a). These results are consistent with the aforementioned findings of an open and closed PGP conformation, where the active site is locked by CP1. Although CP1 undergoes some configurational and conformational changes along the trajectory, as indicated by the 2D-RMSD plots of the ligand in each active site (Fig. 4d), the superimposed snapshots illustrate that CP1 remains at its binding site, occupying the same area as in the reference structure (Fig. 4a, lower right panel).

To clarify which interactions are formed and stably maintained by CP1 and how this inhibitor is supporting the cap-closed PGP state, we measured CP1-PGP interaction distances for putative hydrogen bonds and van der Waals interactions (Supplementary Fig. 14). Interestingly, the crystallographically observed hydrogen bonds of the proximal carboxylic group with Asn199 and Thr215 were not stably maintained. Instead, hydrogen bonds with the side chain atoms of Ser70 and Ser71 were formed after initial fluctuations and moderate conformational changes (Fig. 4c) and observed for almost the entire second half of the trajectory in each monomer (Supplementary Fig. 12 and Supplementary Table 3). The diphenylether moiety engages in continuous interactions with Phe172, leading to distances around 5 Å between the centers of the aromatic rings. The distal benzoate moiety shows some early oscillations in the distances to His175, but rapidly stabilizes in between the side chains of Pro127 and His175, apparently supported by temporary hydrogen-bond formation of the adjacent amide with the side chain of Ser126, as observed in 64–81% of the trajectory.

## CP1 impairs glycolytic flux

To assess the impact of CP1 on central carbon metabolism, we profiled cellular metabolites by untargeted nuclear magnetic resonance (NMR) spectroscopy. As a cellular model system, we used human fibrosarcoma HT1080 cells, a line whose malignant phenotype is driven by an activated N-ras oncogene[27]. HT1080 cells are known to form highly aggressive tumors in murine xenograft models[28], suggesting potential translational relevance[29]. We compared the effects of CP1 with PGP wildtype (PGP-WT, clone D5) or PGP knockout (PGP-KO, clone C2) HT1080 cells generated with the CRISPR/Cas9 system. Strikingly, the majority of the 45 identified metabolites was significantly different in CP1- versus mock-inhibited cells, and in PGP-WT versus PGP-KO cells. Sparse partial least squares discriminant analyses (sPLS-DA; Supplementary Fig. 15) of the metabolomes clearly separated both groups in the two trials, and the ten metabolites that contributed to the decisive principal component 1 were also detected as significantly different in the trial groups by biomarker analysis. Pathway enrichment analyses

and topology scoring indicated significant effects on 'glycolysis and gluconeogenesis', among other pathways (Supplementary Tables 4–7; Supplementary Fig. 16).

Because of the conserved role of PGP for glycolytic flux[12], we next compared the effects of CP1 inhibition and PGP deletion on glycolysis in detail. PGP is known to dephosphorylate the pyruvate kinase side product 2-phospho-L-lactate (2PL), thereby preventing the 2PL-mediated inhibition of phosphofructokinase-2 (PFK-2). Through this mechanism, PGP safeguards the PFK-2-catalyzed formation of fructose-2,6-bisphosphate, an allosteric activator of the key glycolytic regulator phosphofructokinase-1 (PFK-1), thus contributing to the maintenance of glycolytic flux[12] (Fig. 5a). Hence, the loss of PGP activity or -expression is expected to reduce the formation of the glycolytic end products pyruvate and lactate. While 2-PL was not resolved in our NMR spectra, we found significantly reduced pyruvate and lactate levels in both CP1-inhibited and PGP-deficient cells, consistent with a reduced glycolytic flux upon PGP-targeting (Fig. 5b). CP1 treatment (3 h, 33 μM CP1; see Supplementary Note 2 and Supplementary Discussion for an explanation of the required CP1 concentrations) reduced pyruvate levels (−10%), lactate levels (−45%) and acetate levels (−42%; potentially reflecting pyruvate decarboxylation[30]) (Fig. 5b). Since alanine levels were also significantly lower in CP1-inhibited cells (−52% after 3 h; Supplementary Fig. 17), pyruvate levels may be buffered by the gluconeogenic formation of pyruvate from alanine under these conditions. Finally, CP1 treatment or PGP-KO markedly decreased cellular glucose concentrations (−30%, −61%, −74% in cells treated for 3 h with 33 μM CP1, for 24 h with 100 μM CP1, or in PGP-KO cells, respectively; Fig. 5b and Supplementary Fig. 17), in line with the previously reported decrease in the consumption of extracellular glucose observed in PGP-KO HCT116 cells[12]. In sum, these data support the conclusion that CP1 diminishes glycolytic flux, as expected for a PGP inhibitor.

## CP1 impairs PGP-dependent cell proliferation

Glucose consumption and -catabolism, including flux through the pentose phosphate pathway, are intimately connected with proliferative metabolism[31]. Indeed, PGP phosphatase-inactivated mice ($Pgp^{D34N/D34N}$) die in utero with severe growth defects, and murine embryonic fibroblasts obtained by dissociating viable E8.5 $Pgp^{D34N/D34N}$ embryos do not proliferate under standard in vitro culture conditions[13], indicating that PGP activity can influence cell proliferation. To characterize the suitability of CP1 as a tool to modulate cellular PGP functions, and to address the on-target activity of this compound, we next investigated cell proliferation.

The comparison of PGP-WT and PGP-KO HT1080 cells (see Supplementary Fig. 18a for protein blots) showed that PGP deletion reduced the proliferation of single cell clones derived from two different single guide RNAs (gRNAs), or from five pooled gRNAs in a comparable manner (Supplementary Fig. 18b, left and middle panel). CP1 was not cytotoxic at the employed concentration of 100 μM (Supplementary Fig. 18c), but decreased the proliferation of PGP-WT cells to the level of PGP-KO cells, which were insensitive to CP1 treatment (Supplementary Fig. 18b, middle panel). Despite this initial proliferation defect, PGP-KO C2 cells eventually proliferated like their PGP-WT counterparts (Supplementary Fig. 18b, right panel), suggesting that cells can adapt to PGP deficiency (see also Supplementary Note 3). The adaptation of PGP-KO C2 cells reproducibly happened around passage 15, and occurred each time after the PGP-KO C2 cells were put into culture. Nevertheless, long-term PGP-deficient cells remained insensitive to CP1, and CP1 inhibited the proliferation of PGP-WT cells overexpressing wildtype PGP, but not of PGP-WT cells expressing the CP1-insensitive PGP-P127E variant (Supplementary Fig. 18b, right panel).

To better evaluate the on-target activity of CP1, we additionally characterized a CRISPR/Cas9 PGP-knockdown clone (PGP-KD clone

B1-10; ~25% remaining PGP expression; Fig. 5c). Similar to PGP-KO cells, PGP-KD cells proliferated less than their WT counterparts (Fig. 5c). CP1-treatment decreased the proliferation of PGP-WT, but not of PGP-KD cells. The re-expression of PGP-WT in PGP-KD cells normalized their proliferation to the level of PGP-WT cells, and CP1 treatment of these cells again inhibited their proliferation. Importantly, the expression of the enzymatically active, but CP1-resistant PGP variant PGP-P127E (Fig. 3d) in PGP-KD cells reverted their proliferation back to the level of PGP-WT cells, but did not restore responsiveness to CP1 (Fig. 5c). These findings strongly support the conclusion that CP1 mitigates cell proliferation in a PGP-dependent manner.

## CP1 induces PGP-dependent lipid droplet accumulation

Given the PGP-dependent block in glycolytic flux, we finally investigated the impact of CP1 and PGP-KO on a cellular nutrient stress response. Upon energy stress, cells remodel their metabolism and shift from a reliance on glycolysis to fatty acid breakdown for survival[32]. Fatty acids are stored as triacylglycerides (TAGs) in intracellular lipid droplets (LDs)[33,34]. During a cellular energy crisis, the abundance of LDs increases in an adaptive response to buffer lipotoxicity and ER stress, and to increase cellular viability[35,36]. Indeed, we have previously observed that genetic *Pgp*-inactivation or PGP knockdown leads to intracellular TAG and LD build-up in mice, rats, and murine cell lines[10,13,37].

Supplementary Figure 18d shows that CP1 triggered a time- and concentration-dependent accumulation of LDs in HT1080 cells. To test whether CP1-induced LD formation was dependent on PGP inhibition, we re-examined the drug response in PGP-deficient cells. Figure 5d shows that CP1-treatment effectively increased LDs in PGP-WT but not in PGP-KO cells, whose mean basal LD content resembled CP1-treated PGP-WT cells. The re-expression of PGP-WT in PGP-KO cells decreased their LD content to control cell levels, whereas CP1 treatment of these cells again triggered a strong LD accumulation. In contrast, the expression of the CP1-insensitive mutant PGP-P127E in PGP-KO cells reverted the elevated LDs of PGP-KO cells back to WT levels, but did not restore their responsiveness to CP1 (Fig. 5d). In sum, these results strongly argue that the observed LD accumulation was caused by CP1-induced PGP inhibition, and validate CP1 as a pharmacological inhibitor of PGP.

# Discussion

The discovery of metabolite repair proteins has revealed additional layers of control in central carbon metabolism[4,8], but target molecules that can be pharmacologically addressed to modulate pathway flux remain to be defined. Here, we demonstrate the druggability of PGP, a haloacid dehalogenase-type phosphatase and prototypical metabolite repair enzyme that is required to prevent a block in the pentose phosphate pathway and glycolysis[12]. We identify and characterize CP1 as a selective small-molecule PGP inhibitor. Detailed structural analyses reveal that CP1 locks PGP in a catalytically incompetent conformation by targeting a hydrophobic pocket located between the enzyme's core and cap domains. This intriguing mechanism may be broadly applicable to capped HAD phosphatases, and facilitate the future design of highly selective HAD phosphatase inhibitors.

Although phosphatases are considered notoriously difficult to target, the successful development of allosteric protein phosphatase inhibitors is now finally changing this view[38,39]. While only a few mammalian HAD phosphatases have been studied in depth, substantial progress has been made in targeting e.g. EYA2 tyrosine phosphatase in cancer[19,40]. Importantly, HAD phosphatases can show remarkable intrinsic substrate specificity, and do not require an association with regulatory or targeting subunits to impart substrate selectivity[11,26,41]. Inhibiting the activity of a particular HAD phosphatase therefore offers an approach to precisely modulate a specific substrate or small set of

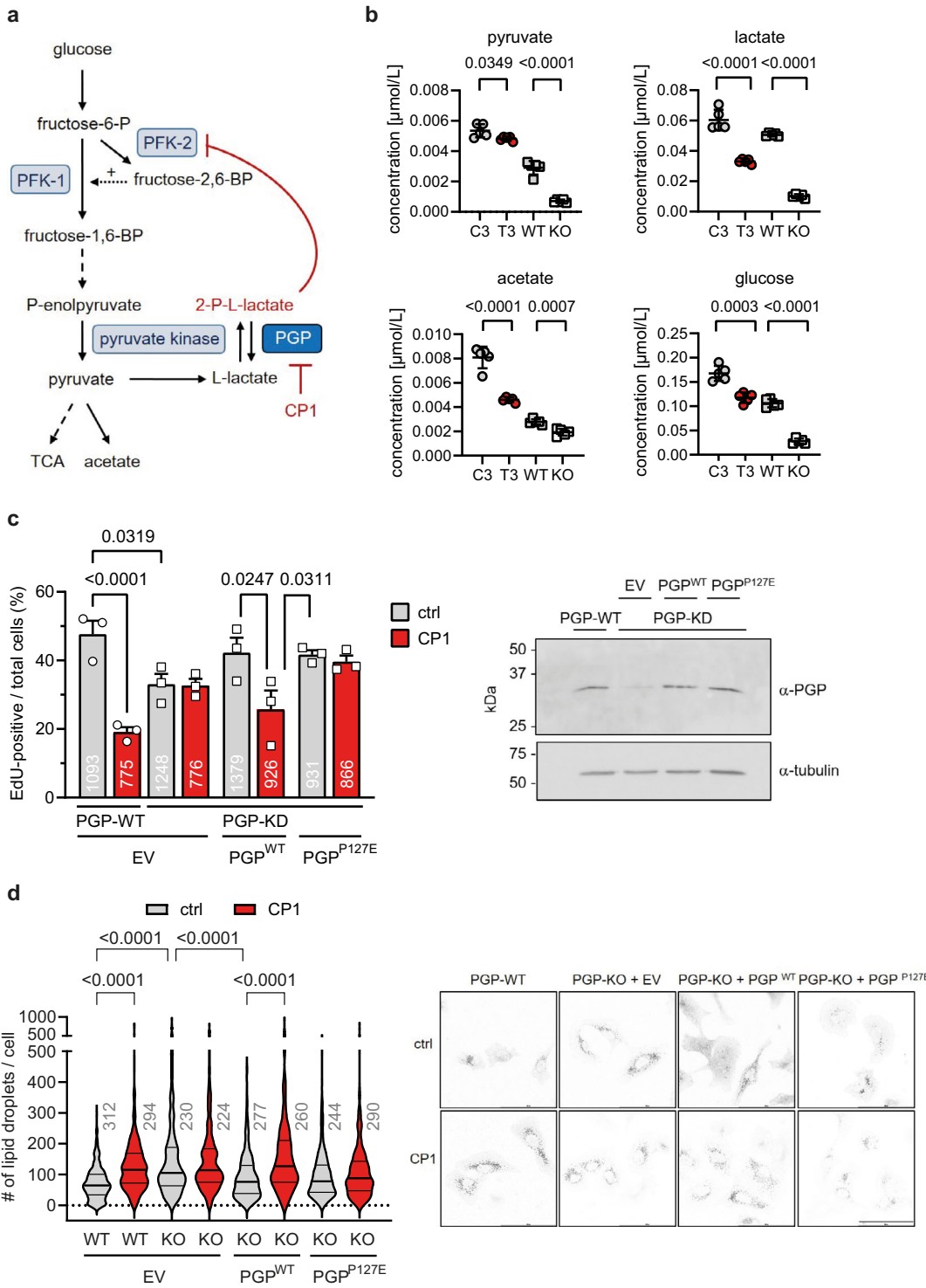

**Fig. 5 | Effects of CP1 in cells. a** Left panel, role of PGP in glycolysis. 2-Phospho-L-lactate (2PL), generated by pyruvate kinase, inhibits PFK2. This mechanism reduces the formation of the PFK1 activator fructose-2,6-bisphosphate, and decreases glycolytic flux. PGP maintains glycolytic flux by dephosphorylating 2PL. PFK1/2, phosphofructokinase-1/2; TCA, tricarboxylic acid cycle. **b** Quantification of cellular metabolites in HT1080 cells by NMR spectroscopy. Cells were treated for 3 h with the DMSO solvent control (C3) or with 33 μM CP1 (T3); PGP-WT (WT, clone D5) or PGP-KO (KO, clone C2) cells are shown for comparison. All data are mean values ± S.D. of $n = 5$ biologically independent experiments. Statistical analysis: two-sided $t$-tests; $p$-values are indicated. **c** Effect of PGP deficiency on cell proliferation. Left, EdU incorporation assays in PGP-WT (clone EV3) or PGP-KD (clone B1-10) HT1080 cells transfected with empty vector (EV), PGP-WT or PGP-P127E. Cells were treated with DMSO (ctrl) or CP1 (100 μM). The total number of scored cells is given

in the bars. Results are mean values ± S.E.M. of $n = 3$ biologically independent experiments. Statistical analysis: ordinary one-way ANOVA with Šidák's multiple comparisons test. Multiplicity-adjusted $P$ values are shown. Right, protein blot of cells analyzed in the left panel. Uncropped blots in Source Data. **d** Analysis of cellular lipid droplets (LDs). Left, quantification of perilipin-3-positive LDs in PGP-WT (clone D5) or PGP-KO (clone C2) HT1080 cells transfected with empty vector (EV), PGP-WT, or PGP-P127E. Cells were incubated for 7 h in the absence (0.1% DMSO control, ctrl) or presence of CP1 (100 μM). The total number of analyzed cells in $n = 3$ biologically independent experiments is indicated. Statistical analysis: ordinary one-way ANOVA with Tukey's multiple comparisons test; multiplicity-adjusted $P$-values < 0.05 are indicated. Violin plots indicate median (middle line) and the two quartiles. Right, representative images of LDs. Shown are inverted grey-scale pictures. Scale bar, 100 μm. Source data are available as Source Data File.

substrates. As such, HAD phosphatases conceptually represent a rich source of pharmacologically largely untapped targets.

CP1 treatment of HT1080 human fibrosarcoma cells recapitulates three previously described effects of PGP loss: a reduction of glycolytic flux[12], a decrease in cell proliferation[12,13], and an increase in lipid droplet formation[13,37]. We therefore anticipate that CP1 will be a useful tool compound to probe the functional consequences of acute PGP inhibition. The broader implication of this work lies in the interest in innovative approaches to target cellular metabolism. Future studies in e.g. a range of cancer cells and suitable disease models will need to evaluate whether the approach of blocking a metabolite repair protein such as PGP to control flux through the pentose phosphate pathway and glycolysis is effective with an adequate therapeutic index. If this turns out to be the case, perhaps more potent PGP inhibitors could be exploited therapeutically to modulate e.g. immune responses in inflammation, autoimmunity or cancer[2].

# Methods

## Materials
Unless otherwise specified, all reagents were of the highest available purity and purchased from Sigma-Aldrich. Compounds **1** (CP1), **1B**, **1C**, **2** (CP2), **2B**, **2 C**, **2D**, **2E** and **3** (CP3) were from ChemDiv. Compounds **1D**, **1E** and **1F** were from Ambinter.

## Phosphatase plasmids and commercially available phosphatases
The following phosphatase constructs (in pCMV6) were purchased from Origene: murine LHPP (NM_029609; catalog #MC211738), murine LPIN1, transcript variant 2 (NM_015763, #MC204460), murine MDP1 (NM_023397, #MR201315), murine NANP/HDHD4 (NM_026086, #MR203153), murine NT5C (NM_015807, #MR202028), human NT5M (NM_020201, #RC207737), murine PNKP (NM_021549, #MR207938), and murine PSPH (NM_133900, #MR202626). Purified human NT5C1A (NM_032526, #TP324617) and purified human PHOSPHO1 (NM_178500, #TP314434) were from Origene. The murine PHOSPHO2 construct (in pFLCI) was from Source BioScience (clone ID: I420023L04). Calf intestinal alkaline phosphatase (CIP) was from NEB. Purified, recombinant human calcineurin (PP2B) and human PTP1B (residues 1-322) were from Millipore (#207005 or #539736, respectively).

## Molecular cloning
The PGP variant *Pgp*$^{D34N; C297S}$ in pETM11 (EMBL) was generated from murine *Pgp*$^{C297S}$ (ref. 14). *Pgp*$^{S71R}$, *Pgp*$^{S71R;F172W}$, *Pgp*$^{P127E}$ and *Pgp*$^{172FDPH>YDEQ}$ in pETM-11 were generated by mutagenizing murine *Pgp*$^{WT}$. S7 Phusion Polymerase was used for *Pgp*$^{S71R}$, *Pgp*$^{S71R;F172W}$; Q5 Hot Start High-Fidelity DNA Polymerase (NEB) was used for all other constructs. *Lhpp*, *Mdp1*, *Nanp*, *Pnkp*, *Phospho2* and *Psph* were subcloned into the *Nco*I (*Pci*I for *Psph*) and *EcoR*I restriction sites of pETM11. The following primers (oligonucleotide sequence 5'– 3'; fwd, forward; rev, reverse) were used:

*Pgp*$^{D34N}$
(fwd: GAGGTGGACACGCTGCTGTTCAACTGCGATGGCGTGCTGTG;
rev: CACAGCACGCCATCGCAGTTGAACAGCAGCGTGTCCACCTC);
*Pgp*$^{S71R}$
(fwd: CACCAACAACAGCAGAAAGACTCGCACGG;
rev: GTGGTTGTTGTCGTCTTTCTGAGCGTGCC);
*Pgp*$^{F172W}$
(fwd: GTAGTGGTGGGCTGGGACCCACACTTCAGC;
rev: CATCACCACCCGACCCTGGGTGTGAAGTCG);
*Pgp*$^{P127E}$
(fwd: CTCGGCTGCTAAGGCCTCGCTGCCCAGCAGGTAGGC;
rev: GCCTACGTGCTGGGCAGCGAGGCCTTAGCAGCCGAG);
*Pgp*$^{172FDPH>172YDEQ}$
(fwd: GTAGCTGAACTGTTCGTCGTAGCCCACCACTACCGCGCG;
rev: GTGGTGGGCTACGACGAACAGTTCAGCTACATGAAGCTC);
*Lhpp*

(fwd: GGGGCCATGGCCGCATGGGCTGAG;
rev: GGGGAATTCTCACTTGTCCGTGTACTT);
*Mdp1*
(fwd: GGGCCATGGCGCGGCTGCCAAAGC;
rev: GGGGAATTCTCAGAGTCCAGCTTGGGC);
*Nanp*
(fwd: GGGCCATGGGGCTGAGTCGGGTCCGCGC;
rev: GGGGAATTCTTACACAGACATGCTTTT);
*Phospho2*
(fwd: CATGCCATGGGAATGAAAGTTCTGTTGGTGT;
rev: TAAGAATGCGGCCGCTCACATCTTTATTAGAAATTGTAA);
*Pnkp*
(fwd: GGGGCCATGGCACAGCTTGGATCCC;
rev: GGGGAATTCTCAGCCCTCGGAAAACTG);
*Psph*
(fwd: GGGACATGTTCTCCCACTCAGAGCTG;
rev: GGGGAATTCTCACTCCTCCAGTTCTCC).
All constructs were verified by sequencing.

## Expression and purification of recombinant phosphatases
All constructs for bacterial protein expression and purification contained an N-terminal His$_6$-tag and were in pETM11 (EMBL). Constructs were transformed into *Escherichia coli* BL21 (DE3) (Stratagene); to increase solubility, proteins were co-expressed with the chaperones groES-groEL-tig from the pG-Tf2 plasmid (Takara Bio Inc.) according to the manufacturer's instructions. All purification steps were carried out at 4 °C.

PGP was expressed for 20 h at 28 °C after induction with 0.5 mM isopropyl β-D-1-thiogalactopyranoside (IPTG). Cells were harvested at 8000 × g for 10 min and resuspended in TNM-1 (50 mM triethanolamine, 200 mM NaCl, 5 mM MgCl$_2$; pH 7.5) supplemented with 10 mM imidazole and protease inhibitors (EDTA-free protease inhibitor tablets; Roche Applied Science). Cells were lysed in the presence of 150 units/mL DNase I (Applichem) using a cell disruptor (Constant Systems), and cell debris was removed by centrifugation (10,000 × g, 30 min, 4 °C). Cleared supernatants were loaded on a HisTrap HP nickel sepharose column (Cytiva) operated on an ÄKTA liquid chromatography system equipped with Unicorn 5.11 software (GE Healthcare) in TNM-1, and His$_6$-tagged proteins were eluted using a linear 10-400 mM imidazole gradient in TNM-1. PGP peak fractions were pooled, and the His$_6$-tag was cleaved with TEV protease for 2–3 days at 4 °C. Subsequently, cleaved PGP was separated from uncleaved PGP and the His-tagged TEV protease on a HisTrap HP nickel sepharose column. Fractions containing untagged PGP were pooled, concentrated (10 kDa MWCO; Amicon Ultra-15, Millipore), further purified on a HiLoad 16/60 Superdex 200 pg size exclusion chromatography column (GE Healthcare) and eluted in TNM-1.

Murine PDXP and the murine PDXP/PGP hybrid protein consisting of the PGP cap domain (amino acids 114–233) fused to the PDXP core domain (amino acids 1–100 and 208–292)[11] were both in pETM11, and expressed and purified as described above for PGP with the following modifications: cultures were incubated for 18 h at 20 °C after induction with IPTG. After centrifugation, cells were resuspended in TNM-2 (100 mM triethanolamine, 500 mM NaCl, 20 mM imidazole, 5 mM MgCl$_2$; pH 7.4). Cleared supernatants were loaded on a HisTrap HP nickel sepharose column in TNM-3 (50 mM triethanolamine, 500 mM NaCl, 20 mM imidazole, 5 mM MgCl$_2$; pH 7.4), and His$_6$-tagged proteins were eluted using a linear gradient of up to 50% TNM-4 (50 mM triethanolamine; 250 mM NaCl; 500 mM imidazole; 5 mM MgCl$_2$; pH 7.4). Peak fractions were pooled, the His$_6$-tag was cleaved, and untagged PDXP or PDXP/PGP was isolated, concentrated and further purified by size exclusion chromatography in TNM-5 (50 mM triethanolamine; 250 mM NaCl; 5 mM MgCl$_2$; pH 7.4).

Murine PGP$^{C297S}$ (ref. 14) was obtained by gene synthesis (GeneArt/Thermo FisherScientific). The D34N mutation was subsequently

introduced into murine PGP$^{C297S}$ by nested PCR, using the *Pgp$^{D34N}$* primers listed above under Molecular Cloning. PGP$^{D34N; C297S}$ was purified and the His$_6$-tag was removed as described above for PDXP protein with the following modifications: cultures were incubated for 20 h at 28 °C, protein overexpression was induced by addition of IPTG to a final concentration of 1 mM, and metal affinity chromatography was performed using a HisTrap excel nickel sepharose column (Cytiva).

LHPP, MDP1, NANP, PNKP and PSPH were expressed for 20 h at 20 °C after induction with 0.2 mM IPTG. Cells were harvested at 8000 × *g* for 15 min and resuspended in TNM-5 supplemented with 10 mM imidazole and 5 μg/mL aprotinin, 5 μg/mL leupeptin, 1 μg/mL pepstatin and 1 mM 4-(2-aminoethyl)-benzene-sulfonyl fluoride (Pefabloc). Cells were lysed by sonication after addition of 1 mg/mL lysozyme and 50 μg/mL DNAse, and cell debris was removed by centrifugation (14,000 × *g*, 20 min, 4 °C). Cleared supernatants were loaded onto TALON metal affinity resin columns (Clontech), and non-specifically bound proteins were removed by washing in TNM-5 supplemented with 10 mM imidazole. His$_6$-tagged proteins were batch-eluted in two successive steps, using 200 and 400 mM imidazole in TNM-5. Fractions containing eluted phosphatases were identified by Coomassie Blue-staining of SDS-PAGE gels, and dialyzed against TNM-5. Protein concentrations were determined with the Micro BCA Protein Assay Kit (Thermo Scientific).

### High-Throughput Screen for PGP Modulators

Chemical library: The FMP small molecule repository used for primary screening consisted of 41,536 compounds in total, divided into subsets with compounds that represent a structural diversity set (33,088 compounds comprising a World Drug Index-derived collection of small molecules structurally related to about 500 different core scaffolds of drugs, according to the concept of maximum-common-substructures[42]; compounds with known pharmacological activity (LOPAC), FDA-approved drugs, and drug candidates which failed in clinical trials (3168 compounds). In addition, the library contains compounds obtained from academic donators world-wide (5280 compounds).

High-throughput screening protocol: Compounds were screened at a final concentration of 10 μM in a total reaction volume of 30 μL in black, non-transparent 384-well assay plates (Corning, #3573). Measurements were conducted in the following assay buffer: Triethanolamine (30 mM), NaCl (30 mM), MgCl$_2$ (5 mM), DTT (1 mM), Triton X-100 (0.01% v/v), pH 7.5. DTT was included as a reducing agent to suppress the redox-dependent formation of PGP oligomers and to ensure maximal PGP activity[14]. For measurements using PGP, final concentrations of 10 μM DiFMUP and 4 ng/μL (0.12 μM) PGP enzyme were used (the $K_M$ value of DiFMUP could not be determined as a saturation could not be observed with DiFMUP concentrations up to 400 μM). For measurements using PDXP, 5 μM DiFMUP and 10 ng/μL (0.3 μM) PDXP were used ($K_M$ ~ 5.1 μM). For both target proteins, screening was therefore conducted at a substrate concentration corresponding to $K_M$ or lower. DiFMUP and target protein solutions were prepared as 3-fold concentrated solutions in assay buffer. Using a microplate dispenser (MicroFlo Select Microplate Dispenser, Biotek), 10 μL of assay buffer were dispensed into columns 1–24 of a 384-well microtiter plate. 0.3 μL of compound solution were transferred from 384-well compound stock plates (containing compounds dissolved to 1 mM in DMSO in columns 1–22 and DMSO in columns 23 and 24) to the assay plates using a liquid handling workstation equipped with a 384-channel pipetting head and disposable tips (Biomek FxP, Beckman Coulter). 10 μL of enzyme solution was dispensed into columns 1–23 of the assay plate, and 10 μL of assay buffer was added to column 24 using a manual multi-pipet. This pipetting scheme created control samples in column 23 with enzyme and DMSO (100% enzyme activity) and control samples in column 24 without enzyme (0% enzyme activity). The plates were then incubated at room temperature for 15 min to allow binding of the compounds to the enzyme. Finally, 10 μL of DiFMUP substrate solution were dispensed to columns 1–24 of the assay plate, the plate shaken and centrifuged with a quick spin to remove air bubbles, and immediately read in a microplate reader (Safire II, Tecan). Fluorescence was measured from the top with an excitation wavelength of 360 nm and emission wavelength at 460 nm (20 nm bandpass each). Five time points were measured over a duration of 9 min. The PerkinElmer LabChip-3000 software was used to generate the .csv flat text files containing the substrate/product ratio.

Concentration-dependent assays: 5 μL of concentrated stock solutions (10 mM in DMSO) of compounds active in primary screening were picked onto a new compound mother plate by the FMP compound management facility, and serial 2-fold inter-plate dilutions were created as follows: using a liquid handler equipped with a 384-channel pipetting head, 5 μL of DMSO were added to the compound mother plate, the samples mixed using the pipetting head, and 5 μL of the diluted compound samples were transferred onto a new compound mother plate. This process was repeated 8 times, thereby creating a set of 9 compound mother-plates with decreasing concentrations. These serially diluted compound plates were then assayed in 2 technical replicates as described above, giving concentration-dependent activity data in the concentration range from 0.2 μM to 50 μM. Because the highest compound concentration used was 50 μM, we defined this value as an IC$_{50}$ cutoff to ensure that the IC$_{50}$ estimate is an interpolation of generated data, according to the recommendations of the NIH Assay Guidance Manual (https://www.ncbi.nlm.nih.gov/books/NBK91994/#assayops.Determination_of_EC50IC50).

Data analysis: Using in-house developed software, the slope (as indicator of the enzyme activity) and intercept (as indicator of compound autofluorescence) were derived from the kinetic trace measured in each well. The slopes from the 100%-activity and 0%-activity control samples of each plate were used to calculate the Z'-factors of the plates as a quality control parameter for the assay. Data of the compound samples were normalized separately for each plate, by calculating Z-scores using all sample wells of a plate, and calculating activities relative to the 100% and 0% control samples of a plate using statistically robust estimators[43]. Final reports and IC$_{50}$/EC$_{50}$ curve fits were then created using the Pipeline Pilot software 9.5 (Biovia).

### Compound solubility and solvent controls

Compound solubilities in 100% DMSO were as follows. CP1: 100 mM; CP2: 15 mM; CP3: 50 mM. **2B**: 10 mM, **2D**: 50 mM, **2E**: 100 mM, **2F**: 50 mM. DMSO solvent control concentrations in the different assays varied depending on the requirements of the respective experiment (see below for further details). However, the DMSO concentration was kept constant across each assay. For SAR experiments, all compound stocks were prepared at 10 mM in 100% DMSO. A constant final DMSO concentration of 0.4% was maintained under all conditions, and solvent control samples contained 0.4% DMSO without compounds.

### IC$_{50}$ determinations, enzyme kinetics, and compound selectivity

Experiments were generally conducted at room temperature (RT) in TMN30 (30 mM triethanolamine, 5 mM MgCl$_2$, 30 mM NaCl; pH 7.5), supplemented with 5 mM DTT and 0.01% (v/v) Triton X-100, with the following exceptions: PHOSPHO1/2 activity was measured in a buffer containing 50 mM imidazole (pH 7.0), 5 mM MgCl$_2$, 0.01% (v/v) Triton X-100 and 5% (v/v) glycerol. The buffer for PHOSPHO1 was further supplemented with 5 mM DTT, and its activity was measured at 37 °C. Calcineurin activity against the PKA regulatory subunit type II (phosphopeptide DLDVPIPGRFDRRVpSVAAE) was assayed in calcineurin assay kit buffer (Millipore #207005); PTP-1B activity against the insulin receptor β-subunit phosphopeptide (amino acids 1142-1153, TRDI-pYETDYYRK) was measured in 150 mM NaCl, 50 mM 2-(N-morpholino) ethanesulfonic acid, 1 mM DTT, 1 mM EDTA, 0.05% (v/v) NP-40; pH 7.2. Calf intestinal alkaline phosphatase activity was assayed in 100 mM

triethanolamine (pH 9.4), 1 mM MgCl$_2$, 20 μM ZnCl$_2$. The 3'-P-oligonucleotide 5'-ATTACGAATGCCCACTCCTC-[PO$_4$$^{3-}$]–3' was used as a PNKP substrate. Purified phosphatases were pre-incubated for 5 min (for compound selectivity assays) or 15 min (for IC$_{50}$ determinations and enzyme kinetics) at RT with serial dilutions of CP1-3. A fixed concentration of 0.1% DMSO was employed for all compounds. Solvent control samples contained 0.1% DMSO without compound. Dephosphorylation reactions were started by the addition of the indicated substrate; buffer without enzyme served as a background control. Prior to compound testing, time courses of inorganic phosphate release from the respective phosphatase substrates were conducted to ensure assay linearity. 2-Phospho-L-lactate was a kind gift of Dr. Guido Bommer; 4-phosphoerythronate was from BLD Pharmatech GmbH. Inorganic phosphate release was detected with malachite green solution (Biomol Green; Enzo Life Sciences); the absorbance at 620 nm (A$_{620}$) was measured on an Envision 2104 multilabel reader, using the Wallac Envision Manager software 1.2 (Perkin Elmer Life Sciences). Glycerol 3-phosphate (G3P) dephosphorylation was measured with the EnzChek Phosphate Assay Kit (Thermo Fisher Scientific), and analyzed at 330/360 nm on a Clariostar multiplate reader and Clariostar software 5.21 R2 (BMG Labtech). Para-nitrophenyl phosphate (pNPP) dephosphorylation to the colored product para-nitrophenol was monitored colorimetrically on an Envision 2104 multilabel reader at A$_{405}$. Released phosphate was determined by converting the values to nmol P$_i$ with a phosphate standard curve. Data were analyzed with GraphPad Prism 7.04. For IC$_{50}$ determinations, log$_{inhibitor}$ versus response was calculated for a Hill slope of −1. To derive $K_M$ and $k_{cat}$ values, data were fitted by nonlinear regression to the Michaelis-Menten equation.

For the analysis of phosphatase activities in cell lysates, LIPIN-1, NT5C and NT5M (myc-DDK tagged) were expressed in HEK-AD293 cells. Cells were cultured as described below, seeded at a density of $0.5 \times 10^6$ cells/3 cm dish overnight, and transfected with *Lpin1, Nt5c, Nt5m* or empty vector control (1.5 μg plasmid each) using TransIT-LT1 (Mirus Bio). The next day, cells were rinsed twice with 1 mL 0.9% NaCl at RT, and scraped in 200 μL ice-cold lysis buffer (TNM30 without DTT, supplemented with 0.2% (v/v) Triton X-100, 5 μg/mL aprotinin, 1 μg/mL pepstatin, 5 μg/mL leupeptin and 1 mM Pefabloc). Cells were lysed by passing through a 25 G × 5/8 needle on ice. Unbroken material and cellular debris were removed by centrifugation (21,000 × $g$, 15 min, 4 °C), and the protein concentration of the clarified lysates was determined using the Micro BCA assay (Pierce). Time courses of P$_i$ release were performed to find linear assay conditions. Phosphatase activities in the absence or presence of the indicated compounds were determined in 5 μL of the respective supernatants (~5 μg cellular proteins), as described above for purified phosphatases. Lysates of vector control transfected cells were used for background correction. All data are mean values of three independent experiments.

### Analysis of CP1 and CP3 combinations on PGP activity

Enzyme inhibition assays were performed as described above, except that murine PGP was incubated in the simultaneous presence of serial CP1 and CP3 dilutions. All wells contained 0.1% DMSO as a solvent control; control wells contained DMSO only. Matrix combination data were processed using the Loewe synergy model and Combenefit software 2.021 (http://sourceforge.net/projects/combenefit/)[44]. Briefly, the software extracts the effects of CP1 and CP3 as single agents from the combination data, which are then fitted with a concentration-response curve. Based on the two single agent concentration-response curves, a combination concentration-response surface is derived. This surface provides a reference for a non-synergistic (additive/independent) combination. The experimental combination concentration-response surface is then compared to the model-generated one, resulting in a synergy distribution in concentration space[44].

### Differential Scanning Fluorimetry

Differential scanning fluorimetry (Thermofluor) assays were performed with PGP$^{WT}$ at varying compound concentrations (244 nM–250 μM for CP1 and CP3, 146 nM–125 μM for CP2) in the presence of 1% DMSO. Solvent control samples contained 1% DMSO without compounds. In a total volume of 24 μL, inhibitor dilutions were incubated for 10 min with 4 μM PGP at RT. The temperature-dependent unfolding was followed in the presence of 1 μL Sypro Orange (1:40 dilution) on a Stratagene Mx3005P thermocycler (Agilent Technologies). Assays were performed as triplicates and the raw unfolding curves were normalized, averaged and fitted to the Boltzmann equation using OriginLab [OriginPro 2019b (64-bit) build 9.6.5.169].

### Isothermal titration calorimetry (ITC)

Purified murine PGP-WT or PGP-P127E was dialyzed overnight against ITC buffer (30 mM HEPES, 30 mM NaCl, 5 mM MgCl$_2$, 1 mM tris(2-carboxyethyl)phosphine/TCEP; pH 7.5). All calorimetric measurements were performed at 25 °C on an ITC200 microcalorimeter (GE Healthcare). Experiments were conducted with a fixed concentration of 2% DMSO. The heat signals released by 15 consecutive 2.4 μL injections of 1 mM CP1 or 0.5 mM CP3 solutions that were titrated into 100 μM or 50 μM PGP, respectively, were measured under continuous mixing at 750 rpm. Both CP1 and CP3 were diluted from a 50 mM stock solution in 100% DMSO, and PGP solutions likewise contained 2% DMSO. To reduce the effects of ligand leakage from the syringe during baseline equilibration, the heat release of a single 1.2 μL injection was recorded prior to the actual titration experiment, and discarded during data analysis with Nitpic 2.0.7[45]. To determine background heats of dilution, control experiments of ligand into buffer, buffer into protein and buffer into buffer titrations were performed.

### Size-exclusion chromatography (SEC)

Purified, recombinant murine PGP-WT or the PGP-C297S variant (30 μM each) were incubated for 20 min at 22 °C in triethanolamine (50 mM), NaCl (250 mM), MgCl$_2$ (5 mM); pH 7.5, supplemented with CP1 (100 μM) or the DMSO solvent control (0.1%). A volume of 900 μL was loaded onto a Superdex 200 10/300GL (Cytiva) size exclusion chromatography column operated on an ÄKTA liquid chromatography system.

### PGP Crystallization and data collection

For the crystallization of full-length PGP, PGP$^{D34N; C297S}$ (30 mg/mL in TMN50, containing 50 mM triethanolamine, 5 mM MgCl$_2$, 250 mM NaCl; pH 7.5) was supplemented with 5 mM unbuffered TCEP. Plate-shaped apo-PGP$^{D34N; C297S}$ crystals were grown at 20 °C in 0.1 M Tris-HCl pH 8.5, 0.2 M (NH$_4$)$_2$SO$_4$, 12% (w/v) PEG 8000 using the sitting-drop vapor diffusion method. For co-crystallization, PGP$^{D34N;C297S}$ (25 mg/mL in TMN50) was supplemented with 5 mM unbuffered TCEP and 1 mM CP1. Prism-shaped plates of CP1-bound PGP$^{D34N;C297S}$ were grown at 20 °C in 0.49 M NaH$_2$PO$_4$ and 0.91 M K$_2$HPO$_4$ using the sitting-drop vapor diffusion method. Crystals were cryoprotected for flash-cooling in liquid nitrogen by soaking in mother liquor containing 25% (v/v) glycerol for apo-PGP$^{D34N;C297S}$ (apo-PGP) or 25% (v/v) PEG 400 for CP1-bound PGP$^{D34N;C297S}$ (CP1-PGP complex). Diffraction data for apo-PGP were collected on the ESRF beamline ID23-1 and for the CP1-PGP complex on the EMBL beamline P13 at the PETRA III synchrotron (DESY, Hamburg). Diffraction data were processed using XDS[46] and further analyzed with Aimless[47] of the CCP4 suite[48]. As Aimless detected significant anisotropy for apo-PGP and the PGP-CP1 complex, data were reprocessed with Staraniso (http://staraniso.globalphasing.org/cgi-bin/staraniso.cgi). The structure of apo-PGP$^{D34N; C297S}$ was solved by molecular replacement with the program Phaser[49] with the structure of the dimeric murine PDXP/PGP hybrid (PDB entry 4BKM) as search model. The structure of the inhibitor complex was solved with the AB-dimer of the apo-PGP structure as search model. Initial real space

refinement and restrained refinements were performed in Refmac[50]. Restrained refinement with the addition of TLS refinement, in the case of the apo-PGP structure, was continued in Phenix[51] and Buster (https://www.globalphasing.com/buster/). A polder electron density map[52] was calculated using Phenix.

## Molecular dynamics simulations

MD simulations were carried out for apo-PGP and the CP1-PGP complex. Starting structures of PGP with wild-type active site were built from the crystal structures of the full-length PGP$^{D34N; C297S}$ (C-D chains, PDB: 7PO7) and CP1-bound PGP$^{D34N; C297S}$ (A-B chains, PDB: 7POE). The Molecular Operating Environment (MOE) 2022.02 (Chemical Computing Group ULC, 1010 Sherbooke St. West, Suite #910, Montreal, QC, Canada, H3A 2R7, 2022) was used for the initial preparation of the structures. The wild-type active site was modelled based on a structural alignment of PGP with crystal structures of PDXP available in the PDB. As PDXP structure 5AES showed the lowest RMSD (1.06 Å) after alignment with the core residues of CP1-PGP$^{D34N; C297S}$, it was used to extract the catalytic $Mg^{2+}$ ion along with two coordinated water molecules and place it into the active site of PGP. A third water molecule was placed at the position where a phosphate oxygen of the PDXP inhibitor in PDB entry 5AES coordinates to the $Mg^{2+}$ ion. Finally, the mutated Asn34 was reverted to the wild-type Asp34, and all residues within 5 Å of the $Mg^{2+}$ ion were subjected to energy minimization with the AMBER14:EHT force field in tether mode to an RMS-gradient of 0.001 kcal/(mol·Å).

CP1 was docked to this structure (as described below) to obtain an energetically favourable starting configuration. The docking results obtained in chain A and chain B differed from the crystallographic binding mode in the CP1-PGP$^{D34N; C297S}$ by only 0.48 Å and 0.73 Å, respectively.

Except for the CP1 inhibitor bound to the active site of each monomer, all other non-protein molecules (i.e., glycerol, potassium ions and water, as well as the CP1 molecules localized outside the active site in the crystal structure) were removed from the system to prepare it for the MD simulation. The protein N- and C-termini were capped with ACE (acetyl, i.e. -COCH₃) and NME (N-methyl, i.e. -NH-CH₃) groups, respectively. The system was protonated at pH 7.5. Histidine residues were defined as HIE tautomers, i.e., with the hydrogen at the epsilon nitrogen. Cys104 and Cys243 were kept in the reduced state (CYS). CP1 was parametrized with AmberTools18 from Amber 2018[53] and the General AMBER Force Field (GAFF)[54] using antechamber and parmchk. The atomic charges were derived from a Restrained Electrostatic Potential (RESP) fit based on electrostatic potentials calculated with Gaussian09[55] at the Hartree-Fock level of theory with the 6−31 G* basis set. PGP parameterization was performed using the AMBER force field ff14SB[56]. After an initial energy minimization of the built structures with a maximum of 2000 cycles through pmemd, the systems were solvated in a TIP3P[57] cuboid water box by means of tleap with a minimum distance of 14.0 Å between solute and periodic box edge. Charge neutrality of the systems was ensured with sodium ions.

All MD simulations were carried out with pmemd of Amber18 under periodic boundary conditions. The systems were thermalized and equilibrated as follows: The solvent was first heated from 100 K to 300 K in the NVT ensemble, using a harmonic potential restraint of 100 kcal/(mol·Å²) on the solute (protein and ligand) during 0.5 ns. The temperature was controlled via Langevin dynamics[58] with a collision frequency of 1 ps⁻¹. Then, the system was cooled to 100 K with a reduced positional restraint of 10 kcal/(mol·Å²) for 0.25 ns. This restraint was further lowered to 1 kcal/(mol·Å²) during the next 0.5 ns, while the system was reheated to 300 K. The relaxation at 300 K was extended for 0.25 ns maintaining the applied restraint of 1 kcal/(mol·Å²). To reach the density of interest, the system was switched to an NPT ensemble, employing a Berendsen barostat[59] at 1 bar and a relaxation pressure time of 5 ps. Under NPT conditions, the

system was further relaxed, conserving the restraint during the first 0.5 ns, followed by 1.0 ns of unrestrained simulation.

After this preparatory phase, a production run of 100 ns was carried out for each system. Electrostatic interactions were handled with the Particle Mesh Ewald method[60], a cut-off of 10 Å was applied for the van der Waals interactions (8 Å during the preparatory phase). Bonds involving hydrogens were constrained using the SHAKE algorithm, allowing a time step of 2 fs in the simulations. The Langevin dynamics were applied with a collision frequency of 3.0 ps⁻¹. The relaxation pressure time was switched to 2 ps at 1 bar. Coordinates of the systems were saved every picosecond. The resulting trajectories were imaged and analyzed using cpptraj from AmberTools18[53]. Structural illustrations from the saved coordinates were prepared with PyMOL 2.4.1[61].

Principal component analysis of the MD simulations was carried out with utilities from Bio3D[42]. The underlying PDB format trajectory for each principal component was obtained from Bio3D and postprocessed with PyMOL 2.4.1 to generate movies from two different views. The individual files were then assembled and labelled with Shotcut (www.shotcut.org, version 22.06.03, Meltytech, LLC). The same colour scheme as in the corresponding PCA figures in Supplementary Fig. 13 was applied.

## Docking

To investigate the putative binding mode of the inhibitors CP1-3 and the substrate 2-phosphoglycolate (PG) to the wild-type active-site of PGP, computational docking studies were carried out with Genetic Optimisation for Ligand Docking (GOLD) 2022.1.0[62]. Protein and ligand structures were prepared using MOE 2022.02. The wild-type active-site modelled from the CP1-PGP crystal structure as described above under Molecular Dynamics Simulations was used for docking studies at both active sites (i.e, AS-A of monomer A and AS-B of monomer B), using ChemPLP and ASP as scoring functions within GOLD. The search radius for the binding site was set to 10 Å around CP1 as reference structure; for PG, the search area was centred at the $Mg^{2+}$ ion with a radius of 10 Å. Water molecules coordinated to the $Mg^{2+}$ ion were considered in three different modes: off, toggle, and on, with a maximal translation distance of 1 Å. Parameters of the genetic algorithm were set as follows: population size 750; number of operations 500,000; crossover 90; mutation 95; and migration 25. For each scoring function and water setting 100 docking runs were carried out, resulting in 200 docking poses per water mode and a total of 600 docking poses per binding site. All poses were rescored with DrugScoreX (DSX)$^{CSD}$ 0.9[63] and structurally clustered with Fconv 1.24[64] using an RMSD cut-off of 2 Å. PyMOL 2.4.1 was used for visual inspection.

## Cell culture, transfection and CRISPR/Cas9 engineering

Human embryonic kidney HEK-AD293 cells (Stratagene) and human HT-1080 cells (a malignant fibrosarcoma line established from a treatment-naïve patient; LGC Standards) were cultured in Dulbecco's modified Eagle's medium (DMEM; 4.5 g/L glucose) supplemented with 10% (v/v) heat-inactivated fetal calf serum (FCS; Gibco), 2 mM L-glutamine, 100 U/mL penicillin and 100 μg/mL streptomycin (complete DMEM). Media and cell culture supplements were from Thermo Fisher Scientific.

Guide RNA (gRNA) sequences were taken from the human GeCKOv2 library[65] and cloned into pLentiCRISPRv2. The following primers were used:

A1 PGP, HGLibA_36275, 5′-CGGAAGCCGGTTGTCCATGT-3′;
A2 PGP, HGLibA_36276, 5′-ACGCTGCTGTTCGACTGCGA-3′;
A3 PGP, HGLibA_36277, 5′-CAACCCCGAGCGCACCGTCA-3′;
B1 PGP, HGLibB_36229, 5′-GCGACGACGCCCGCTGCGTG-3′;
B3 PGP, HGLibB_36230, 5′-CGCACGGACCCGCGATGAAG-3′.

pLentiCRISPRv2 was digested using BsmBI (NEB); oligonucleotides were phosphorylated and annealed with T4-PNK (NEB) and ligated with T4 ligase. To generate PGP-WT control, PGP-KO or PGP-KD

clones, parental HT1080 cells were transfected with either the empty vector (pLentiCRISPRv2), the single vectors containing one of the five different PGP gRNAs, or the five pooled vectors containing PGP-directed gRNAs, using Lipofectamine 2000.

## Western blotting

Cultured cells were lysed in 50 mM Tris, pH 7.5; 150 mM NaCl; 1% (v/v) Triton X-100; 0.5% (w/v) sodium deoxycholate; 0.1% (w/v) sodium dodecyl sulfate; 5 µg/mL aprotinin, 1 µg/mL leupeptin, 1 µg/mL pepstatin, and 1 mM Pefabloc. After clarification of the lysates by centrifugation, protein concentrations were determined using the Micro BCA Protein Assay Kit. Proteins were separated by SDS-PAGE and transferred to nitrocellulose membranes by semidry-blotting. Antibodies were purchased from the following providers: Merck Millipore (α-actin mAb1501, dilution 1:5000); Cell Signaling Technology (α-tubulin clone DM1A, dilution 1:2000; α-GAPDH clone 14C10, dilution 1:1000); Santa Cruz Biotechnology (α-PGP clone E10, dilution 1:75). Rabbit polyclonal α-PGP antibodies (employed at 1:1000 dilution) were generated by Charles River using purified, untagged full-length murine PGP$^{D34N}$ as an immunogen in New Zealand White rabbits, and antibodies were purified by affinity chromatography[11].

## NMR spectroscopy, data normalization and statistical analysis

Parental HT1080 cells or HT1080 CRISPR/Cas9_PGP-WT (clone D5) or CRISPR/Cas9_PGP-KO (clone C2) cells were cultured as described above. For NMR experiments, 2.7 million cells were seeded in 10 cm dishes for all samples. The next day, PGP inhibition was performed using either 33 µM CP1 over 3 h, or 100 µM CP1 over 24 h; the same amount of pure DMSO solvent was added in parallel to the respective control cells. To ensure comparable conditions between CP1-treated and CRISPR/Cas9 HT1080 lines, the latter cells were also grown for a total of 48 h and treated with 0.1% DMSO for 24 h. For each condition, six samples were prepared; one of which was used to count the number of cells at the end of the sample preparation. Free metabolites were obtained by methanol extraction: After washing the samples with ice-cold PBS, they were placed in liquid nitrogen for 2 min, followed by mixing the samples cooled on ice with 5 mL methanol. The samples were mixed for 15 min at 4 °C, and centrifuged at 2.5 × $g$ for 30 min at 4 °C. The solvent was removed by evaporation with a N$_2$ gas flow over 5 h on ice, followed by freeze-drying under vacuum. The remaining pellet was dissolved in 700 µL D$_2$O, and transferred to a 5 mm borosilicate NMR tube. NMR experiments were performed with a BRUKER AVANCE NEO 600 spectrometer operating at 600.13 MHz for $^1$H detection equipped with a cryogenic NMR probe offering enhanced sensitivity. The $^1$H NMR spectra were acquired using a double watergate pulse sequence with excitation sculpting for water suppression. A relaxation delay of 4 s was sufficient. The π and π/2 pulses performed with a radio frequency power of 62.5 kHz were combined with 2 ms selective rectangular shape pulses optimized for efficient suppression of the H$_2$O $^1$H resonance, which was chosen as the transmitter frequency. The two blocks of gradient pulses were conducted over 1 ms per pulse using gradients of 0.155 T/m and 0.055 T/m with a smoothed rectangular shape profile (SMSQ10.100). Signal averaging was achieved over 512 scans using an acquisition time of 1.38 s. Acquisition and pre-processing of NMR spectra was performed under the control of a workstation with TopSpin 4.0.7. Data processing included Fourier transformation, zero order phase correction, 1 Hz line broadening using a Lorentzian window function and automatic baseline correction with a polynomial of fifth degree. A 9 mmol/L aqueous sucrose solution was used as a calibration compound to calibrate an ERETIC2 reference for absolute concentration determination[66,67]. Based on the sucrose reference, an electronic signal was placed in each $^1$H spectrum to calibrate the deconvolution software Chenomx NMR suite. A standard protocol for quantitative NMR was applied: The signal corresponding to the calibration compound was integrated and normalized according to

the number of protons of this compound. The unknown concentration of compound x in the presence of the calibration compound was calculated according to:

$$C_x = \frac{I_x}{I_{cal}} \frac{N_{cal}}{N_x} C_{cal}$$

With I, N, and C being the integral area, number of nuclei, or concentration of the calibration (*cal*) or the targeted compound (*x*), respectively. Due to spectral superposition, glycerol-3-phosphate could not be quantified in $^1$H NMR spectra. Therefore, we conducted $^{31}$P - $^1$H HMBC 2D experiments and quantified glycerol-3-phosphate using the phosphorylcholine peak and the respective concentration known from $^1$H NMR spectra. Even though signal intensity in such experiments also depends on $^{31}$P-$^1$H J-couplings, we assume that the error caused hereby is insignificant, especially as we aimed to investigate relative concentration differences caused by cell treatments. $^1$H and $^{31}$P pulses were performed at radio frequency powers of 62.5 kHz and 22.2 kHz, respectively. Three gradient pulses were used (sequence hmbcgpndqf) over 1 ms each with gradient strengths of 0.06 T/m, 0.06 T/m and 0.05 T/m. The relaxation delay was set to 2 s, the acquisition time to 0.328 s. 128 experiments were acquired for each 2D experiment using a time increment of 204 µs and averaging over 16 scans. Data processing was achieved as described above except for the Qsine window function in both dimensions. All measurements were performed at 7 °C. Raw datasets are provided on Figshare under the accession code [https://doi.org/10.6084/m9.figshare.21316533][68].

Data were normalized by dividing the concentration of each metabolite by the cell number and by the absolute metabolite content of the respective sample; this value was multiplied with the group average of the absolute metabolite content. This approach removes effects caused by cell number differences, while maintaining those generated by the trial conditions. Multivariant data analysis was performed using custom-made R scripts employing the mixOmics package[69]. For sPLS-DA, the data were additionally centered. Classic univariate receiver operating characteristic (ROC) curve analysis used to calculated AUC values and pathway enrichment analysis was performed using MetaboAnalyst[70] using the web server MetaboAnalyst 5.0 (https://www.metaboanalyst.ca/MetaboAnalyst/ModuleView.xhtml). The pathway analysis procedure employed the options Global Test and relative betweenness centrality in combination with the KEGG *Homo sapiens* database.

## Cell viability assays

HT1080 cells were seeded onto 96-well plates (5000 cells per well), and treated for 24 h with the indicated CP1 concentrations or with the appropriate DMSO solvent control. Cell viability was assessed using AquaBluer according to the manufacturer's recommendations (Multi-Target Pharmaceuticals, LLC). Fluorescence (excitation: 540 nm; emission: 590 nm) was analyzed on a Clariostar microplater reader.

## Cell proliferation assays

Parental HT1080 cells or the indicated CRISPR/Cas9-engineered control (PGP-WT) or PGP-KO cells were grown in complete DMEM. For the expression of PGP-WT or PGP-P127E, cells were transfected with either the pcDNA3 empty vector control or the indicated constructs in pcDNA3 using X-tremeGENE (Roche/Merck), according to the manufacturer's instructions. The next day, cells were seeded at a density of 2000 cells per well of a 96-well microtiter plate, and incubated with 100 µM CP1 or 0.1% DMSO for 16 h. To assess DNA synthesis, cells were labeled for 2.5 h with 10 µM 5-ethynyl-2'-deoxyuridine (EdU), fixed and permeabilized according to the manufacturer's recommendations (Click-iT EdU Alexa Fluor 488 imaging kit, Invitrogen/Thermo Fisher). Cells were counterstained with 4',6-diamidine-2-phenylindole (DAPI), embedded with Fluoromount-G (Thermo Fisher), and imaged on a

Nikon TE Eclipse epifluorescence microscope equipped with a 4 × objective and NIS elements software AR 5.02.00. EdU and/or DAPI-positive cells were analyzed using the ImagePro-Plus software version 7.0 (Media Cybernetics). Statistical significance was assessed using GraphPad Prism 9.4.1 software.

## Lipid droplet staining

HT1080 cells were seeded on glass coverslips (15,000 cells per well of a 12-well plate) in complete DMEM, and incubated for 16 h with 100 μM CP1 or with 0.1% of the DMSO solvent control. Cells were fixed for 15 min with 4% para-formaldehyde, permeabilized and blocked with 0.2% saponin/2% bovine serum albumin (BSA) for 1 hr, and immunostained for 1 hr using α-rabbit TIP47/perilipin 3 antibodies (1:100 dilution, abcam #47638) in 0.2% saponin/1% BSA. Primary antibodies were detected with Alexa Fluor 488-labeled goat anti-rabbit secondary antibodies (1:400; Molecular Probes/Thermo Fisher Scientific), and nuclei were counterstained with DAPI. Cells were mounted with Immu-Mount (Fisher Scientific), and imaged by fluorescence microscopy on a THUNDER imager (THUNDER Imager Live Cell & 3D Cell Culture & 3D Assay system, Leica DMi8; Leica Microsystems), using a 63 × objective and the Leica Application suite X software 3.7.4.23463. To determine the number of lipid droplets per cell, cell borders were defined semi-automatically, and bright objects in this region of interest were counted using the ImagePro-Plus software version 7.0. Statistical significance was assessed using GraphPad Prism 9.4.1 software.

## Mass spectrometry and quantification of intracellular CP1

HT1080 CRISPR/Cas9_PGP-WT (clone D5; PGP-WT) or CRISPR/Cas9_PGP-KO (clone C2; PGP-KO) cells were cultured as described above. As specified below, cells were analyzed in parallel at passage 5 and at passage 15. For all samples, cells were seeded in 10 cm dishes and grown overnight. The next day, PGP-WT or PGP-KO cells (-1 million cells/dish) were harvested by trypsinization. For this, cells were rinsed in PBS (37 °C), trypsinized, suspended in complete DMEM (37 °C), sedimented at $300 \times g$, 37 °C for 4 min, washed once in PBS (37 °C), and sedimented again at $300 \times g$, 37 °C for 4 min. Cell pellets were snap-frozen in liquid nitrogen.

For the determination of intracellular CP1 concentrations, PGP-WT cells (clone D5, passage 5) were treated with 33 μM CP1 over 3 h or with 100 μM CP1 over 7 h. Cells (-1 million cells/dish) were rinsed in PBS (37 °C), trypsinized, sedimented at $300 \times g$, 4 °C for 4 min, washed once in PBS (4 °C), and sedimented again at $300 \times g$, 4 °C for 4 min. Cell pellets were snap-frozen in liquid nitrogen.

Water-soluble metabolites were extracted with 50 μl ice-cold MeOH/$H_2O$ (80/20, v/v) containing 0.01 μM lamivudine. After centrifugation of the resulting homogenates, supernatants were transferred to a RP18 SPE (50 mg/1 mL tubes; Phenomenex) that had been activated with 0.5 mL $CH_3CN$ and conditioned with 0.5 mL of MeOH/$H_2O$ (80/20, v/v). The eluate of the RP18 SPE-column was evaporated in a SpeedVac (Savant, Thermo Fisher Scientific). Dry sample extracts were redissolved in 75 μL 5 mM $NH_4OAc$ in $CH_3CN$/$H_2O$ (50/50, v/v). Twenty μL of supernatant were transferred to LC-vials. Metabolites were analyzed by LC-MS using the following settings: 3 μL of each sample was applied to an XBridge Premier BEH Amide (2.5 μm particles, 100 × 2.1 mm) UPLC-column (Waters). Metabolites were separated with Solvent A, consisting of 5 mM $NH_4OAc$ in $CH_3CN$/$H_2O$ (40/60, v/v) and solvent B consisting of 5 mM $NH_4OAc$ in $CH_3CN$/$H_2O$ (95/5, v/v) at a flow rate of 200 μL/min at 45 °C by LC using a DIONEX Ultimate 3000 UHPLC system (Thermo Fisher Scientific). A linear gradient starting after 2 min with 100% solvent B decreasing to 10% solvent B within 23 min, followed by 16 min 10% solvent B and a linear increase to 100% solvent B in 2 min was applied. Recalibration of the column was achieved by a 7 min prerun with 100% solvent B before each injection. Ultrapure $H_2O$ was obtained from a Millipore water purification system (Milli-Q Merck Millipore). HPLC-MS solvents, LC-MS $NH_4OAc$,

standards and reference compounds were purchased from Merck. All MS-analyses were performed on a high-resolution Q Exactive mass spectrometer equipped with a HESI probe (Thermo Fisher Scientific) in alternating positive- and negative full MS mode with a scan range of 69.0–1000 m/z at 70 K resolution and the following ESI source parameters: Sheath gas: 30, auxiliary gas: 1, sweep gas: 0, aux gas heater temperature: 120 °C, spray voltage: 3 kV, capillary temperature: 320 °C, S-lens RF level: 50. XIC generation and signal quantitation was performed using TraceFinder v3.3 (Thermo Fisher Scientific) integrating peaks which corresponded to the calculated monoisotopic metabolite masses (MIM +/− H + ± 3 mMU). Analyses of PGP-WT and PGP-KO cells were performed in $n = 5$ biologically independent experiments. Analyses of PGP-WT cells for the determination of intracellular CP1 levels were performed in $n = 3$ biologically independent experiments. The cellular concentration of CP1 was estimated based on the number of counted cells, and 1 pL as the typical single cell volume of a mammalian tissue culture cell (https://bionumbers.hms.harvard.edu).

## Reporting summary

Further information on research design is available in the Nature Research Reporting Summary linked to this article.

## Data availability

The previously published PDB entries 4BKM of a PDXP/PGP hybrid protein, and 5AES of PDXP in complex with an inhibitor are used in this manuscript. X-ray crystallographic data generated in this study have been deposited in the PDB and can be accessed as follows: apo-PGP$^{D34N; C297S}$: PDB entry 7PO7, and CP1-bound PGP$^{D34N; C297S}$: PDB entry 7POE. Source data are provided with this paper. The source data of molecular dynamics simulations and docking that were generated in this study are provided as part of the Source Data file. The NMR source data generated in this study are available in the Figshare database under accession code [https://doi.org/10.6084/m9.figshare.21316533].

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

## Acknowledgements

We thank Carola Seyffahrt and Nicole Bader for excellent technical assistance, and the ESRF and PETRA III staff at beamlines ID23-1 and P13, respectively. We gratefully acknowledge the expert technical assistance of Martina Fischer. We thank Drs. Carsten Hoffmann and Julia Drube for kindly providing the original PGP CRISPR constructs, and Dr. Guido Bommer for the generous gift of 2-phospho-L-lactate. This work was partly funded by the Deutsche Forschungsgemeinschaft (SFB688 to A.G., FZ82 to H.S.). D.E. was supported by a predoctoral fellowship from the Medical Faculty of the University of Würzburg (Graduate School of Life Sciences). N.Y.C. was supported by a fellowship from the DAAD (German Academic Exchange Service). Moreover, institutional financial support from the Ministerium für Kultur und Wissenschaft des Landes Nordrhein-Westfalen, the Regierender Bürgermeister von Berlin, Senatskanzlei Wissenschaft und Forschung and the Bundesministerium für Bildung und Forschung (01KU1216I) to the Leibniz-Institut für analytische Wissenschaften-ISAS is gratefully acknowledged. This publication was supported by the Open Access Publication Fund of the University of Würzburg.

## Author contributions

A.G. developed the concept and designed the study. M.N. performed the screening campaign and analyzed data; M.N. and J.P.v.K. provided resources. E.J. validated hits and designed, performed and analyzed biochemical experiments with the help of K.H., S.F., D.Y.-G., T.F., D.E., A.Ke., A.Ka. and A.G. Crystallization screens were conducted by J.S. and S.F.; J.S. collected and analyzed data together with H.S., and performed and analyzed ITC and thermofluor experiments. K.H. performed all cell biological experiments, and analyzed data together with A.G. Molecular dynamics simulations and docking experiments were conducted by N.Y.-C.; data were analyzed together with C.S. NMR experiments were conducted by M.A. and R.Ho.; R.Ho. analyzed and interpreted data together with R.He. and A.G.; R.He. provided resources. W.S. performed and analyzed mass spectrometry experiments. A.G. wrote the main paper, H.S. edited and wrote the crystallography part, C.S. wrote the molecular dynamics simulations and docking parts. H.S. and A.G. acquired funding. E.J., R.He., C.S., H.S. and A.G. supervised the work.

## Funding

## Competing interests

The University of Würzburg has filed a patent application relating to the use of compounds described in this work for the treatment of cancer (patent application pending), with A.G. named as inventor. The other authors declare no competing interests.
