## [Peer Review File · Nature Communications]

REVIEWER COMMENTS

Reviewer #1 (Remarks to the Author):

The manuscript reports a very exhaustive study that highlighted three inhibitors - with CP1 being more potent than CP2 and 3 - of PGP, which could find important applications in medical research, especially for immune system disorder and cancers. The amount and the depth of the experiments performed clearly demonstrate the robustness of the study.

However, majors amendments are expected for the NMR spectroscopy methodology part, i.e. more details are needed to ensure that the experiment is detailed enough to be reproduced:

- acquisition parameters
- (pre)-treatment parameters with the software used
- more details are needed also on the quantification of the 45 metabolites, especially because the watergate pulse sequence is not a sequence adapted for absolute quantification
- more details are also requested for the quantification of the glycerol-3-phosphate as 2D NMR experiments are not quantitative without the applications of specific conditions and calibration curve

Added to this, here are some minor suggestions to improve the manuscript:

- in the introduction, more details and context would be expected related to the methods used, rather than highlighting only the results
- Lines 111 to 113 report results on the cap/core domains, although those terms are explained and discussed only in the following section. Some re-ordering might improve the understanding of the reader.
- SAR: why analogs of CP3 have not been explored? An explanation would be appreciated
- SI Fig S5: what is the interest of the loading 2 plots, as there are no separations observed along with the principal component 2 of both models.

Reviewer #2 (Remarks to the Author):

Here the authors set out to develop selective inhibitors for phosphoglycolate phosphatase, PGP, a metabolite repair hydrolase that is a member of the HAD family, and is important for maintenance of efficient glycolytic flux through hydrolysis of 2-phosphoglycolate (PG), 4-phosphoerythronate, an inhibitor of the pentose phosphate pathway, and 2-phospholactate, a byproduct of pyruvate kinase activity that inhibits 2-PFK, with the long term goal of using such an inhibitor as a drug to treat diseases such as chronic inflammation, which are heavily dependent on the glycolytic process and the pentose phosphate pathway. For this purpose, they screened the FMP ~50,000 compound library for PGP inhibitors using DiFMUP as a universal substrate, and the closely-related PDXP pyridoxal phosphate HAD family phosphatase as a counterscreen, and obtained three hits with submicromolar IC₅₀'s, CP1-3, which were shown to be selective for PGP versus PDXP and several additional HAD family hydrolases and other phosphatases. Using classical kinetic analysis, they characterized the mechanism of PGP inhibition by CP1-3 showing that CP1 is partially competitive, whereas CP2 and CP3 are competitive. Next, they carried out apo-PGP dimer crystal and CP1/PGP co-crystal structure studies, establishing that CP1 binds in a pocket adjacent to the active site, and that bound CP1 locks PGP in a closed, inactive conformation with the selectivity cap and flap domains tightly associated, precluding substrate binding. On this basis, they concluded that CP1 acts as an allosteric inhibitor of PGP. The binding mode of CP1 was validated by making and analyzing CP1 contact residue mutants, with Pro127Glu PGP being almost totally resistant to CP1 inhibition. They then went on to test the effects of CP1 treatment in cultured cells, using untargeted metabolite NMR analysis to show that treatment of HT1080 fibrosarcoma cells with 30 μ M CP1 led to changes in the levels of 45 metabolites, with ten of these being shown to form a decisive principal component that distinguished the CP1-inhibited metabolome. Importantly, decreased levels of pyruvate and

lactate were observed in PGP KO and CP1-treated cells, consistent with PGP inhibition reducing glycolytic flux. Using WT and PGP knockout HT1080 cells, combined with re-expression of WT, catalytically-dead, and P127E CP1-resistant mutant PGP, they found that the metabolic effects of CP1 PGP inhibition resulted in decreased proliferation in a manner dependent on PGP. Finally, they showed that CP1 treatment induced formation of lipid droplets, as a stress response to perturbed metabolism, also dependent on PGP inhibition.

These are elegant and convincing studies that employed a wide range of biochemical, biophysical, structural and cell biological approaches to identify and validate CP1 as a selective, cell-permeant inhibitor of PGP activity that can reduce glycolytic flux in cultured cells and induce a lipid droplet stress response. These properties suggest that CP1 could be useful pharmacologically, although this will have to be further tested in in vivo models. In this regard, by many measures CP1 is not a very potent inhibitor, and there is some question how easy it would be to obtain inhibitory CP1 concentrations in vivo.

Points: 1. Introduction: Is PGP the only metabolite repair enzyme that acts on 2-PG, or are there any functional overlaps with the nine other metabolic repair enzymes in the glycolysis pathway?

2. Figure 1: What is the solubility of the CP1-3 compounds? Was DMSO concentration maintained across the assays or varied according to individual compound solubility? If it was varied, were the corresponding PGP-DMSO controls (without compounds) employed in the assay?

3. Can the 2-phosphoglycolate in vitro assay be extended to other known metabolites that PGP repairs, e.g., 2-phospho-L-lactate and 4-phosphoerythronate? Such assays with and without CP1 could support the potency of CP1 against the known cellular substrates of PGP.

4. Do 2B or 2D have better solubility compared to CP2? Are they good leads in comparison to CP1?

5. Why did the steady-state kinetic values, especially K_M 's, change for PGP in the absence of CP1-3 in Supplementary Table S1? They are not within the error range of each other. Is this because of different DMSO concentrations being used for solubilizing CP1-3?

6. Figure 2: Representing the core and cap domains in two different colors would make it much easier for the reader to visualize the changes in chain A vs chain E.

7. Figure 3: Can CP3, which is significantly "smaller", i.e. has one less ring, be modeled into the same pocket as CP1? Do CP1 and CP3 compete for binding to PGP? Are their inhibitory effects on PGP activity additive? If 2-PG is modeled into the active site, does it clash with CP1?

8. Figure 3: Was CP1 binding to P127E PGP measured by ITC?

9. Can the differences in movements in cap domain residues and flap residues be attributed to crystal contacts? A figure to illustrate the arrangement of the dimers in an asymmetric unit might be helpful.

10. If one compares Supplementary Tables S5 and S6, the glycolysis and gluconeogenesis pathways show the highest alteration in rate in PGP-WT vs PGP-KO but it is in fifth place in CP1- vs mock-inhibited cells. Though the pathways listed above have some connection to glycolysis and gluconeogenesis, this leads one to wonder if CP1 is exerting off-target effects by binding to some other proteins involved in these pathways, or is there an alternative explanation for this observation? In terms of whether CP1 might have targets other than PGP in the cell, is it possible to put a handle on CP1 (possibly based on the co-crystal structure) and then do MS analysis of proteins that are captured on immobilized CP1?

11. Figure S7B: By eventually adapting to PGP deficiency, the PGP-KO C2 cell line proliferated like WT. How are 2-phospholactate levels reduced in these cells? Are there any alternative PG hydrolases that could be upregulated and substitute for PGP in its absence? How quickly does adaption happen in terms of number of cell divisions, and does it happen reproducibly every time PGP-KO C2 are put in culture?

12. What is intracellular concentration of CP1 when cells are treated with 30 μ M CP1 (can CP1 be detected by untargeted NMR?) and does it reach PGP IC50 levels?

13. Page 14: What are the determinative 10 metabolites whose levels are affected by CP1 treatment, and are these changes explained by failure to dephosphorylate known PGP substrate molecules?

Minor points: 1. Figure 1A: It is really difficult to pick out the O and N atoms in the CP1-3 structures, even when they are expanded.

Reviewer #3 (Remarks to the Author):

Jeanclous and coworkers reported the identification and extensive biochemical, structural and functional evaluation of CP1, a specific small-molecule PGP inhibitor that blocks PGP-dependent glycolytic flux. Furthermore, the conducted research validated a novel pharmacological target class. Overall, manuscript is written in a clear and detailed way. Please find my comments below:

1) The authors should provide more details about the FMP small molecule repository. If compounds are obtained from the following link : <https://www.leibniz-fmp.de/de/the-screening-unit/downloads>, the authors should include details about how this library is produced.

2. 23 inhibitors that were screened out need to be shown in the supplementary data section.

3. In Figure S1. the total # of compounds binding to PDXP and the total number of ligands that bind to PGP should be detailed.

4. A citation for the IC50 cutoff of 50uM IC50 is required for the initial fluorescence screening.

5. Authors stated that "CP1-3 were also able to block PG dephosphorylation catalyzed by an artificial protein consisting of the PDXP catalytic core and the PGP specificity-determining cap (PDXP/PGP hybrid 11)" ; Does this sentence conflict with the previous sentence stating: "We confirmed that CP1-3 did not inhibit PDXP activity towards pyridoxal 5'-phosphate (PLP)"?

6. The melting points of the inhibitor-PGP complexes aren't correlated with the binding affinities of the compounds. Authors need to discuss this in detail.

7. In Figure S.2, panel A, 1F is defined as an isomeric fragment however, 1E is not labeled.

8. In Figure S.2, panel B, IC50 of CP1 is given as 0.38uM in Figure 1, panel A, it is given as 0,39 and 0,37, "M.m" and "H.s.", respectively.

9. Authors wrote that "In contrast, the CP2 analogs with the exception of 2E still retained

inhibitory potency, with 2B being even slightly more potent than the parental compound.” This data is not provided, please include in the SI.

10. It is needed to test CP1 against NT5C1A with the same concentration used for PGP inhibition.

11. Open and closed conformations of PGP are crucial, their superimposition should be (i.e. chains A-C, F and H) represented in a better way in Figure 2.

12. In line 221, the expression “excellent” should be removed from the sentence.

13. In Figure3B, the Asp34 should be represented and labeled at the figure.

14. Chain-based RMSD values for CP1-bound PGP versus apo-PGP should be given.

15. Use RMSF plots to depict the fluctuations of the residues during molecular dynamics simulations.

16. Together with the Figure 4.A, which isn’t clearly represented, preparation of a simulation movie could help readers to better understand the structural movement during the simulations.

17. In Figure 4.A, representing the domains (cap and catalytic domain) by an arrow, will help with clarification.

18. Authors should also conduct PCA for the PGP simulations.

19. Authors should also conduct binding free energy calculations (GBSA or PBSA) throughout the simulations.

20. In order to analyze the effect of individual binding energy contribution of the binding pocket residues on CP1 binding conduct per residue interaction analysis.

21. To better understand the effect of the active residues to the binding of CP1, the in vitro mutational studies can be repeated by in silico simulations.

22. Further discussion for the use of larger concentrations (100uM) in PGP-dependent cell proliferation and PGP-dependent droplet accumulation assay is needed.

23. Has CP1 previously been used on another target protein, if yes, it would need to be discussed and the activity should be linked to PGP inhibition.

24. Along with CP1; CP2 and -3 should also be used for MD simulations.

25. Furthermore, together with CP1-3 simulations, a negative control simulation with weak PGP binders “among initially tested” could be performed.

26. MD simulations were carried out PGPD34N; C297S. Simulations should also be repeated for wild-type PGP.

27. The system was protonated at pH 8.5 to imitate crystallization conditions, however authors need to test the effect of protonation states at physiological conditions.

28. The authors can expand their SAR study by searching diphenylether and diphenylmethylene dibenzoic acid derivatives in a more diverse library such as ZINC to obtain more hits or they may conduct this search at the FDA library for drug repurposing effort.

29. CP1 binding to the second binding site could be tested in silico.

30. In line 950, correct "Van-der-Waals" to "van der Waals".

31. Sequence identity between murine and human PGP should be added.

Reviewer #4 (Remarks to the Author):

This manuscript reports the targeting of phosphoglycolate phosphatase (PGP) which is an important repair enzyme in glycolytic process. The study is novel in targeting a metabolite repair enzyme.. The manuscript details an exhaustive and well-performed series of experiments using a broad tool-kit of techniques which ultimately support the idea that, since HAD phosphatases can show remarkable intrinsic substrate specificity and do not require an association with regulatory or targeting subunits to impart substrate selectivity, they represent a viable focus for future inhibitor development.

Overall I recommend publication of this work however, the following points should be considered in revision of the manuscript.

Table in Fig. 1 requires units.

The reporting of thermodynamic parameters in the text and as a Table in Fig.1 is redundant without further analysis. For example, provision of the change in entropy data from ITC experiments for the two compounds does not mean anything without any reference to how these data describe the binding event, i.e. do they correlate with entropic features such as removal of water molecules or conformational change of the protein. In the absence of a meaningful structure/thermodynamic analysis the dissociation constant is the only value worth reporting.

The authors report that PGP exists in dynamic equilibrium between homodimers and tetramers. This could have significant implications in the lead compound binding studies. What has been done to mitigate against the possible impact of self-association of the enzymes. For example, is self-association enhanced by compound binding? Is there likely to be any allosteric effect between monomers of ligand binding?

The crystal structural work is very well presented, however the authors report that "Towards the end of the refinement, residual density could be tentatively interpreted with a second CP1 molecule, which is binding to each monomer on the opposite side of PGP. The binding site is quite remote from the active site (closest distance of ~ 15 Å) and is in close proximity to the non-crystallographic twofold axis of symmetry. As the ITC data indicate binding of a single copy of CP1 to each monomer and the crystallographic evidence documents that the second CP1 molecule is weakly bound, no biological significance was attributed to the second binding site." The assumption that, because the second binding event is not visible in the ITC data, it does not exist is incorrect. Some binding events have a zero (or negligible) change in enthalpy at a given temperature. Therefore, strictly speaking to be sure that the second binding event is weak or absent the ITC experiment should be done at a range of temperatures. I would not advocate this in this work, but some mention of the limitations of the experiment should be made.

The cell-based assays focus on the human fibrosarcoma cell line, HT1080. The authors should provide justification for the choice of cell line. Clearly, the demonstration of activity of the compound across a range of cancer cells would provide convincing evidence for the efficacy of targeting of the phosphatase system. This should be mentioned as a limitation of the study.

After the initial screens and binding study the study focuses on CP1, despite the observation that CP2 and CP3 have similar PGP activity on PG and similar affinities for the target. The authors might want to comment on whether the other compounds could be investigated further.

We thank all four reviewers for their constructive comments and suggestions, which have helped us to further improve our manuscript. We have addressed all points raised by the reviewers, and have conducted additional experiments. All new results are fully consistent with our previous observations and conclusions. Please find our point-by-point responses below.

Reviewer #1 (Remarks to the Author):

The manuscript reports a very exhaustive study that highlighted three inhibitors - with CP1 being more potent than CP2 and 3 - of PGP, which could find important applications in medical research, especially for immune system disorder and cancers. The amount and the depth of the experiments performed clearly demonstrate the robustness of the study.

However, major amendments are expected for the NMR spectroscopy methodology part, i.e. more details are needed to ensure that the experiment is detailed enough to be reproduced:

- acquisition parameters

- (pre)-treatment parameters with the software used

We thank the reviewer for this suggestion. We have added the following details regarding acquisition and (pre)-treatment parameters and software used in the Methods section:

“A relaxation delay of 4 s was sufficient. The π and $\pi/2$ pulses performed with a radio-frequency power of 62.5 kHz were combined with 2 ms selective rectangular shape pulses optimized for efficient suppression of the H₂O ¹H resonance, which was chosen as the transmitter frequency. The two blocks of gradient pulses were conducted over 1 ms per pulse, using gradients of 0.155 T/m and 0.055 T/m with a smoothed rectangular shape profile (SMSQ10.100). Signal averaging was achieved over 512 scans using an acquisition time of 1.38 s. Data processing included Fourier transformation, zero order phase correction, 1 Hz line broadening using a Lorentzian window function and automatic baseline correction with a polynomial of fifth degree. A 9 mmol/L aqueous sucrose solution was used to calibrate an ERETIC2 reference for quantification⁶¹. Based on the sucrose reference, an electronic signal was placed in each ¹H spectrum to calibrate the deconvolution software Chenomx NMR suite.”

- more details are needed also on the quantification of the 45 metabolites, especially because the watergate pulse sequence is not a sequence adapted for absolute quantification

The reviewer is also right concerning the objection that spectra with Watergate suppression are not necessarily quantitative. Indeed, we observed that NMR signals close to the water resonance are partially suppressed, for instance those caused by the proton at the anomeric carbon atom of glucose. However, each of those molecules had resonances far from the H₂O signal, which were well suited for quantification.

- more details are also requested for the quantification of the glycerol-3-phosphate as 2D NMR experiments are not quantitative without the applications of specific conditions and calibration curve.

We added experimental details in the Methods section as follows:

“¹H and ³¹P pulses were performed at radio frequency powers of 62.5 kHz and 22.2 kHz, respectively. Three gradient pulses were used (sequence hmbcgpndqf) over 1 ms each with gradient strengths of 0.06 T/m, 0.06 T/m and 0.05 T/m. The relaxation delay was set to 2 s, the acquisition time to 0.328 s. 128 experiments were acquired for each 2D experiment using a time increment of 204 μ s and averaging over 16 scans. Data processing was achieved as described above except for the Qsine window function in both dimensions.”

We are also aware that it is difficult to obtain quantitative results via calibration using only the known HMBC resonance of phosphorylcholine, but as already described in the manuscript, we assume that at least relative concentrations will be correct, since there is no reason why relative signal response between the glycerol-3-phosphate and the phosphorylcholine signal should vary from sample to sample.

Added to this, here are some minor suggestions to improve the manuscript:

- in the introduction, more details and context would be expected related to the methods used, rather than highlighting only the results

We thank the reviewer for this suggestion. During revision of the manuscript, we have included additional methods such as docking studies and mass spectrometry to address the points raised by the reviewers. Like for all other methods, we have provided in-depth details and context on their use throughout the Results, the Methods and the extensive Supplementary Information sections. In view of the number and diversity of the employed methods and the given space constraints, we cannot elaborate more on this in the Introduction.

- Lines 111 to 113 report results on the cap/core domains, although those terms are explained and discussed only in the following section. Some re-ordering might improve the understanding of the reader.

The following sentence was introduced in the Introduction: “PGP consists of a catalytic core harboring a Rossmannoid fold and a large cap domain restricting access to the active site¹¹.”

- SAR: why analogs of CP3 have not been explored? An explanation would be appreciated
CP3 indeed shows higher binding affinity to PGP than CP1 (KD 0.43 μ M for CP3 versus 0.78 μ M for CP1), but is somewhat less potent (IC50 0.81 μ M for CP3 versus 0.37 μ M for CP1). Because CP1 and CP2 are structurally very similar, we restricted our limited SAR analysis to analogs of these two leads in the present study. Regardless, CP3 may be an interesting lead for future studies (please also see Supplementary Discussion).

- SI Fig S5: what is the interest of the loading 2 plots, as there are no separations observed along with the principal component 2 of both models.

Indeed, the values of the second loading plots are not relevant for the separation. It is only shown for completeness as numerous readers might be used to 2D Scores plots with a full description of both components.

Reviewer #2 (Remarks to the Author):

Here the authors set out to develop selective inhibitors for phosphoglycolate phosphatase, PGP, a metabolite repair hydrolase that is a member of the HAD family, and is important for maintenance of efficient glycolytic flux through hydrolysis of 2-phosphoglycolate (PG), 4-phosphoerythronate, an inhibitor of the pentose phosphate pathway, and 2-phospholactate, a byproduct of pyruvate kinase activity that inhibits 2-PFK, with the long term goal of using such an inhibitor as a drug to treat diseases such as chronic inflammation, which are heavily dependent on the glycolytic process and the pentose phosphate pathway. For this purpose, they screened the FMP ~50,000 compound library for PGP inhibitors using DiFMUP as a universal substrate, and the closely-related PDXP pyridoxal phosphate HAD family phosphatase as a counterscreen, and obtained three hits with submicromolar IC50-values, CP1-3, which were shown to be selective for PGP versus PDXP and several additional HAD family hydrolases and other phosphatases. Using classical kinetic analysis, they characterized the mechanism of PGP inhibition by CP1-3 showing that CP1 is partially competitive, whereas CP2 and CP3 are competitive. Next, they carried out apo-PGP dimer crystal and CP1/PGP co-crystal structure studies, establishing that CP1 binds in a pocket adjacent to the active site, and that bound CP1 locks PGP in a closed, inactive conformation with the selectivity cap and flap domains tightly associated, precluding substrate binding. On this basis, they concluded that CP1 acts as an allosteric inhibitor of PGP. The binding mode of CP1 was validated by making and analyzing CP1 contact residue mutants, with Pro127Glu PGP being almost totally resistant to CP1 inhibition. They then went on to test the effects of CP1 treatment in cultured cells, using untargeted metabolite NMR analysis to show that treatment of HT1080 fibrosarcoma cells with 30 μ M CP1 led to changes in the levels of 45 metabolites, with ten of these being shown to form a decisive principal component that distinguished the CP1-inhibited metabolome. Importantly, decreased levels of pyruvate and lactate were observed in PGP KO and CP1-treated cells, consistent with PGP inhibition reducing glycolytic flux. Using WT and PGP knockout HT1080 cells, combined with re-expression of WT, catalytically-dead, and P127E

CP1-resistant mutant PGP, they found that the metabolic effects of CP1 PGP inhibition resulted in decreased proliferation in a manner dependent on PGP. Finally, they showed that CP1 treatment induced formation of lipid droplets, as a stress response to perturbed metabolism, also dependent on PGP inhibition.

These are elegant and convincing studies that employed a wide range of biochemical, biophysical, structural and cell biological approaches to identify and validate CP1 as a selective, cell-permeant inhibitor of PGP activity that can reduce glycolytic flux in cultured cells and induce a lipid droplet stress response. These properties suggest that CP1 could be useful pharmacologically, although this will have to be further tested in *in vivo* models. In this regard, by many measures CP1 is not a very potent inhibitor, and there is some question how easy it would be to obtain inhibitory CP1 concentrations *in vivo*.

Points: 1. Introduction: Is PGP the only metabolite repair enzyme that acts on 2-PG, or are there any functional overlaps with the nine other metabolic repair enzymes in the glycolysis pathway?

We thank the reviewer for raising this important question. We have added the following text to the Introduction (lines 72-75): “PGP is the only mammalian enzyme that has been described to act on 2-phosphoglycolate, 2-phospho-L-lactate and 4-phosphoerythronate, and there are no known functional overlaps with other metabolite repair processes in the context of glycolysis^{8,12}”. Further details are provided in the Supplementary Notes of the revised manuscript.

2. Figure 1: What is the solubility of the CP1-3 compounds? Was DMSO concentration maintained across the assays or varied according to individual compound solubility? If it was varied, were the corresponding PGP-DMSO controls (without compounds) employed in the assay?

CP1 is soluble at 100 mM in 100% DMSO; CP2 at 15 mM in 100% DMSO, and CP3 at 50 mM in 100% DMSO. DMSO solvent control concentrations in the different assays varied depending on the requirements of the respective experiment. However, the DMSO concentration was kept constant across each assay, specifically when different compounds were compared. We have added a paragraph “Compound solubility and solvent controls” in the revised Methods section, and have specified the respective DMSO concentrations in the Methods part or Legends of all assays.

3. Can the 2-phosphoglycolate *in vitro* assay be extended to other known metabolites that PGP repairs, e.g., 2-phospho-L-lactate and 4-phosphoerythronate? Such assays with and without CP1 could support the potency of CP1 against the known cellular substrates of PGP.

As suggested, we have extended the *in vitro* phosphatase assay to 2-phospho-L-lactate and 4-phosphoerythronate. We found that CP1 is equally effective in inhibiting PGP-mediated dephosphorylation of 2-phosphoglycolate, 2-phospho-L-lactate and 4-phosphoerythronate. These new data are now included in the revised table in Fig. 1A, and we show the inhibition curves of 2-phospho-L-lactate, 4-phosphoerythronate and glycerol-3-phosphate in Supplementary Fig. 1.

4. Do 2B or 2D have better solubility compared to CP2? Are they good leads in comparison to CP1? The solubilities of CP2 and its analogs in 100% DMSO are as follows. CP2: 15 mM, 2B: 10 mM, 2D: 50 mM, 2E: 100 mM, 2F: 50 mM. This information is now included in the paragraph “Compound solubility and solvent controls” in the revised Methods section.

2B (IC₅₀ 1.07 μM; maximal inhibition 99%) has a poorer solubility in DMSO than CP2, but is somewhat more active (CP2: IC₅₀ 1.62 μM; maximal inhibition 90%), whereas 2D (IC₅₀ 1.71 μM; maximal inhibition 93%) has a better solubility in DMSO than CP2, but is similar to CP2 in potency and efficacy. We conclude that 2B may be an interesting lead, but that CP1 still appears superior (IC₅₀ 0.37 μM, maximal inhibition 97%, solubility in DMSO 100 mM). However, we have not conducted additional experiments with 2B to substantiate this assumption. We now also mention 2B as a potentially interesting lead in the Supplementary Discussion.

5. Why did the steady-state kinetic values, especially K_M-values, change for PGP in the absence of CP1-3 in Supplementary Table S1? They are not within the error range of each other. Is this because of different DMSO concentrations being used for solubilizing CP1-3?

We thank the reviewer for pointing this out. In this particular experiment, the (very low) DMSO concentrations were kept constant, but in fact varied slightly between CP1 and CP2/CP3. We now describe this in the Table legend (Supplementary Table 2 in the revised manuscript): “DMSO concentrations were kept constant, but varied between CP1 (0.02% DMSO under all conditions) and CP2/CP3 (0.04% DMSO under all conditions). Solvent control samples contained 0.02 or 0.04% DMSO without compound.”

We have compared the K_M values of the three independent experiments (in the absence of CP1-3) using unpaired t -tests, and did not find statistically relevant differences:

	K_M (CP1)	K_M (CP2)	K_M (CP3)
Exp 1	371.5	327.5	445.2
Exp 2	327.3	539.2	361.5
Exp 3	356.0	390.7	575.4

Unpaired t -test K_M (CP1) vs. K_M (CP2): $p=0.3513$

Unpaired t -test K_M (CP1) vs. K_M (CP3): $p=0.1612$

Unpaired t -test K_M (CP2) vs. K_M (CP3): $p=0.6626$

6. Figure 2: Representing the core and cap domains in two different colors would make it much easier for the reader to visualize the changes in chain A vs chain E.

We have changed the colors of cap and core domains in both chains in the revised figure.

7. Figure 3: Can CP3, which is significantly smaller, i.e. has one less ring, be modeled into the same pocket as CP1? Do CP1 and CP3 compete for binding to PGP? Are their inhibitory effects on PGP activity additive? If 2-PG is modeled into the active site, does it clash with CP1?

We have carried out docking studies for CP1, CP2, CP3, and 2-PG using the crystal structure of the CP1-PGP^{D34N;C297S} complex after removal of CP1 and restoration of the wild-type active site (with D34 and Mg²⁺ ion). A detailed description of this modelling and docking study is provided as Supplementary Information. - To briefly answer the questions of the reviewer, CP3 can be well modelled into the same pocket as CP1 (Supplementary Figs. 6, 7). The docking solutions of 2-PG show that it can bind to the active site and adopt favourable conformations without clashing with CP1 (Supplementary Fig. 8).

We have also performed matrix combination assays to analyze PGP activity in the presence of both CP1 and CP3. Data were processed with the Combenefit Software. As shown in Supplementary Fig. 9 of the revised manuscript, no antagonistic or synergistic effects were observed. Consistent with the modeling results, CP1 and CP3 thus inhibited PGP in an additive/independent manner.

8. Figure 3: Was CP1 binding to P127E PGP measured by ITC?

We have now performed the suggested ITC experiment, and found an approximately 15-fold higher KD (12 μ M) for CP1 with PGP-P127E than with PGP-WT. This result is in line with the higher IC₅₀ of CP1 for PGP-P127E compared to PGP-WT (Fig. 3D), and further supports our functional findings demonstrating that PGP-P127E is a phosphatase-competent, but CP1-insensitive PGP variant. The new ITC result is described in lines 290-294 of the revised manuscript.

9. Can the differences in movements in cap domain residues and flap residues be attributed to crystal contacts? A figure to illustrate the arrangement of the dimers in an asymmetric unit might be helpful. As suggested, we have included a figure (Supplementary Figure 4) of the arrangement of the four dimers in the asymmetric unit, which is in this case (space group P1) of course equivalent to the unit, hence crystal contacts are formed solely by unit cell translations. Supplementary Fig. 4 shows that the four dimers are arranged via a pseudo-twofold non crystallographic symmetry axis (oriented along the viewing direction) with the two monomers representing the conformational outliers (monomers D and E) not following the non-crystallographic twofold, i.e. they reside in the “top” two dimers. A further analysis of the packing environments of the cap domains indicates that these two monomers and the A monomer are less engaged in crystal contact than the cap domains of the remaining monomers. As both conformational states are observed when minimal packing constraints are present, we feel

confident to state that the observed conformational changes may at best be induced by indirect crystal contacts.

10. If one compares Supplementary Tables S5 and S6, the glycolysis and gluconeogenesis pathways show the highest alteration in rate in PGP-WT vs PGP-KO but it is in fifth place in CP1- vs mock-inhibited cells. Though the pathways listed above have some connection to glycolysis and gluconeogenesis, this leads one to wonder if CP1 is exerting off-target effects by binding to some other proteins involved in these pathways, or is there an alternative explanation for this observation? We cannot exclude metabolic ‘off-target’ effects of CP1 at this point. However, CP1 treatment or genetic PGP deletion generally altered the same metabolites, albeit often to different extents (which is likely due to the persistent, complete loss of PGP activity in PGP-KO cells *versus* a transient inhibition of PGP activity in CP1-treated cells). Because the *p*-values of the pathway analyses are assigned according to the number of altered metabolites in a given pathway and according to the separation of the respective metabolite *concentrations* in control and trial groups, quantitative metabolite changes affect the resulting rank orders shown in Supplementary Tables 5 and 6 (Supplementary Tables 6 and 7 in the revised manuscript). Regardless, the *p*-values describing the impact of the intervention on glycolysis / gluconeogenesis are highly significant in both cases (1.29E-09 for PGP-KO cells and 3.29E-06 for CP1-treated cells).

In terms of whether CP1 might have targets other than PGP in the cell, is it possible to put a handle on CP1 (possibly based on the co-crystal structure) and then do MS analysis of proteins that are captured on immobilized CP1?

We thank the reviewer for this suggestion. It might be possible to place a chemical handle on CP1 to perform the suggested experiment. However, because CP1 is buried in a cavity that extends from the active site of monomer A towards the dimerization interface with monomer B, we are concerned that steric and charge repulsion effects of this handle might impair CP1-PGP binding and/or affect PGP-PGP homodimer interactions. The feasibility of such an approach will also depend on the ability to immobilize CP1 on a pulldown matrix via a suitable spacer, without restricting CP1 motional degrees of freedom and without losing its interaction with PGP. While this approach is certainly an interesting, yet challenging, option to explore in future studies, we feel that such screens are beyond the scope of the present manuscript.

11. Figure S7B: By eventually adapting to PGP deficiency, the PGP-KO C2 cell line proliferated like WT. How are 2-phospholactate levels reduced in these cells?

2-Phospholactate was not visible by NMR. We have therefore conducted mass spectrometric analyses to measure 2-phospho-L-lactate and 4-phosphoerythronate levels in PGP-KO C2 cells cultured for shorter (passage 5) or longer periods (passage 15; the time at which these cells adapt to PGP deficiency). These results are described in the Supplementary Notes of the revised manuscript. Briefly, we found that 4-phosphoerythronate levels were ~46-fold higher in PGP-KO compared to PGP-WT cells at passage 5, but only ~14-fold higher at passage 15. Thus, HT1080 cells can partially compensate for PGP loss over time. Unfortunately, we were unable to separate 2-phospho-L-lactate from various isomeric metabolites, and can therefore not assess possible changes in 2-phospho-L-lactate levels over time. Regardless, these new results reveal how HT1080 cells can partially compensate for PGP loss over time.

Are there any alternative PG hydrolases that could be upregulated and substitute for PGP in its absence?

We are not aware of any alternative 2-phospho-L-lactate or 4-phosphoerythronate hydrolases that could be upregulated and substitute for PGP in its absence, although this is certainly a possibility in addition to metabolic rewiring processes that lessen cellular dependence of glycolysis.

How quickly does adaptation happen in terms of number of cell divisions, and does it happen reproducibly every time PGP-KO C2 are put in culture?

We have edited the manuscript to clarify these points (lines 406-407): “The adaptation of PGP-KO C2 cells reproducibly happened around passage 15, and occurred each time after PGP-KO C2 the cells were put into culture.”

Given an approximate HT1080 cell doubling time of 30 h (<https://www.dsmz.de/collection/catalogue/details/culture/ACC-315>) and three subcultures per week, 15 passages are reached after 5 weeks. This time frame numerically corresponds to 28 cell divisions.

12. What is intracellular concentration of CP1 when cells are treated with 30 μM CP1 (can CP1 be detected by untargeted NMR?) and does it reach PGP IC50 levels?

NMR is not sensitive enough to detect CP1. To answer this question, we have performed mass spectrometric experiments (see Supplementary Methods). As detailed in the Supplementary Notes and the Supplementary Discussion, we estimated intracellular CP1 concentrations to 0.13 μM or 0.23 μM upon incubation with 33 or 100 μM CP1, respectively. These estimates are slightly below, but in any case in the range of PGP IC50 levels (CP1 IC50 is 0.39 μM for human recombinant PGP in vitro; Fig. 1A). Intracellular CP1 concentrations may nevertheless well reach PGP IC50 levels for the following reasons: (1) It is unknown if the potency of CP1 for recombinant PGP in vitro is identical to its potency towards endogenously expressed PGP. (2) The effective distribution volume for small molecules in the cytoplasm is lower than the total cell volume. (3) The average concentration of a compound in a cell lysate is unlikely to reflect its local subcellular concentrations. (4) Importantly, we observed that intracellular CP1 concentrations are not linearly correlated with the extracellularly applied CP1 concentrations. Based on the results with 33 μM CP1, the theoretical intracellular CP1 concentration after incubation with 100 μM CP1 should be 0.38 μM , not 0.23 μM . Hence, we detected 39% less CP1 than expected. These data suggest that CP1 is intracellularly metabolized or degraded. Because only intact CP1 could be quantified (nothing is known about the intracellular metabolism or degradation products of this compound), we conclude that the intracellularly effective CP1 concentrations are likely underestimated.

13. Page 14: What are the determinative 10 metabolites whose levels are affected by CP1 treatment, and are these changes explained by failure to dephosphorylate known PGP substrate molecules?

The ten metabolites contributing to component one in the 2D Scores plot are listed in the Loading 1 plot of the sPLS-DA, shown in the upper right panel of Supplementary Fig. S13 (phosphocreatine, alanine, aspartic acid, phosphorylcholine, glutamic acid, creatine, glucose, taurine, lysine and -to a minor extent- myo-inositol). To illustrate how these changes may be explained by a failure to dephosphorylate known PGP substrates, we have included a metabolic pathway scheme as Supplementary Fig. 14 in the revised manuscript. In fact, alterations in 9 of the 10 determinative metabolites (with the exception of phosphorylcholine) can be linked to decreased glycolytic flux due to increased levels of the PGP substrate 2-phospho-L-lactate. The CP1-mediated decrease in the levels of phosphorylcholine (the UDP-choline precursor in phosphocholine biosynthesis) may be caused by PGP-dependent alterations in glycerolipid metabolism (refs. 10, 37), and is in line with the reduction of phosphocholine levels observed in PGP-deficient mouse embryos (ref. 13). The methods employed in our current study are not appropriate to address the connections between PGP inhibition by CP1 and metabolite changes in greater detail. For this purpose, we will consider e.g. isotopic tracing experiments in future studies.

Minor points: 1. Figure 1A: It is really difficult to pick out the O and N atoms in the CP1-3 structures, even when they are expanded.

Corrected.

Reviewer #3 (Remarks to the Author):

Jeanclos and coworkers reported the identification and extensive biochemical, structural and functional evaluation of CP1, a specific small-molecule PGP inhibitor that blocks PGP-dependent glycolytic flux. Furthermore, the conducted research validated a novel pharmacological target class. Overall, manuscript is written in a clear and detailed way. Please find my comments below:

Before replying point-by-point to the suggestions made by reviewer #3 with respect to the MD simulations, we would like to thank the reviewer for comment number 26, in which simulations of wild-type PGP were suggested. We have run such simulations in which the wild-type active site was restored and have used them to replace the formerly presented simulations where the active site carried the inactivating D34N mutation and no Mg^{2+} ion. With these new simulations, we can at the same time illustrate the dynamics of the system in a physiologically more relevant state and demonstrate

(combined with docking calculations) that the binding mode of CP1 obtained with the inactivated mutant is essentially maintained in the active form of the enzyme. Moreover, the findings of the original manuscript regarding the overall dynamics of the system remain largely valid.

Upon revision, we also realized that the crystal structures of apo-PGP and CP1-PGP differ in a disulfide bond. While Cys104 forms a disulfide bridge with Cys243 in the apo crystal structure, no such disulfide bond is formed in the CP1-PGP crystal structure. This is most likely due to the different crystallization conditions, with a somewhat lower amount of TCEP in relation to the protein concentration in case of the apo-PGP crystals. The original simulations had been started from the crystal structures without properly defining the state of Cys104 and Cys243 in the apo simulation, which may have led to erroneous results. In the new simulations carried out with the wild-type active site, both in the apo and in the complexed form, we have consistently set Cys104 and Cys243 to the reduced state, i.e., without disulfide bond, which is considered to be the physiologically more relevant state.

Any changes in the text are highlighted in the revised manuscript. Obviously, all figures related to MD simulations were newly prepared with the data from the simulations of PGP with wild-type active site. Further details are mentioned in the point-by-point response below.

1) The authors should provide more details about the FMP small molecule repository. If compounds are obtained from the following link (<https://www.leibniz-fmp.de/de/the-screening-unit/downloads>) the authors should include details about how this library is produced.

We have included further specifications and a reference in the Supplementary Methods. On a legal basis, we cannot provide structural information to all compounds as commercial vendors forbid this transfer to third party users. Only for the CBB1 sub-collection of 16.544 compounds we are allowed to provide a structure data file. To protect intellectual property from academic donations we provide structural data only with permission of the chemists who donated their compounds.

2. 23 inhibitors that were screened out need to be shown in the supplementary data section. We now show these compounds in SI Table S1.

3. In Figure S1, the total # of compounds binding to PDXP and the total number of ligands that bind to PGP should be detailed.

We have added these details to the revised Supplementary Figure 1.

4. A citation for the IC₅₀ cutoff of 50uM IC₅₀ is required for the initial fluorescence screening. We have now explained the chosen IC₅₀ cutoff and have added a citation in the Supplementary Methods section.

5. Authors stated that “CP1-3 were also able to block PG dephosphorylation catalyzed by an artificial protein consisting of the PDXP catalytic core and the PGP specificity-determining cap (PDXP/PGP hybrid”. Does this sentence conflict with the previous sentence stating: “We confirmed that CP1-3 did not inhibit PDXP activity towards pyridoxal 5-phosphate (PLP)”?

The statements in the two sentences do not conflict, but rather point to the importance of the PGP cap domain for CP1-3 binding. The catalytic activity and specificity of HAD-type phosphatases depend on the coordinated action of the catalytic core domain and the specificity-determining cap domain (ref. 15). In previous work, we have created a PDXP/PGP hybrid protein consisting of the PDXP core domain fused to the PGP cap domain (ref. 11). This artificial protein is unable to dephosphorylate PLP (ref. 11), but effectively dephosphorylates PG (this work), demonstrating that residues in the PGP cap are required for PG coordination and hence specific substrate recognition.

The table in Fig. 1A shows that CP1-3 did not inhibit PDXP-mediated PLP dephosphorylation. In contrast, CP1-3 inhibited PG dephosphorylation catalyzed by PGP or by the PDXP/PGP hybrid protein. Therefore, we conclude that “These data suggest that CP1-3 interact with the PGP cap domain ...” (lines 120-121).

6. The melting points of the inhibitor-PGP complexes aren't correlated with the binding affinities of the compounds. Authors need to discuss this in detail.

As stated in the manuscript we were unable to conduct ITC measurements with CP-2. For the remaining compounds the changes in melting temperature qualitatively agree with the binding affinities measured by ITC; CP-1 with a K_D of 0.8 μM showed a smaller stabilization (T_M change of 4.6 $^\circ\text{C}$) compared to CP-3 with a K_D of 0.4 μM and a T_M change of 7.1 $^\circ\text{C}$). Due to the fact that the data points at high concentrations of CP-3 deviate from the fitted curve, the 7.1 $^\circ\text{C}$ value should be taken with a grain of salt. As stated in the manuscript our main objective of these measurements was to ensure that inhibitor binding does not lead to protein destabilization, which we believe to be fully supported by the data.

7. In Figure S.2, panel A, 1F is defined as an isomeric fragment however, 1E is not labeled. The revised text now also refers to 1E, and reads: "We tested a pyrrolidinedione-derivative (1B), two isomers (1C and 1D) and two isomeric fragments (1E and 1F) of CP1..." (lines 140-141).

8. In Figure S.2, panel B, IC_{50} of CP1 is given as 0.38 μM in Figure 1, panel A, it is given as 0.39 and 0.37 μM , respectively.

The IC_{50} of CP1 in the SI Figure S2B for murine PGP is $0.38 \pm 0.08 \mu\text{M}$. In independent sets of experiments, we determined the IC_{50} of CP1 in Figure 1, panel A as $0.39 \pm 0.03 \mu\text{M}$ for human PGP, and as $0.37 \pm 0.03 \mu\text{M}$ for murine PGP. These results clearly attest to the high reproducibility of CP1-mediated inhibition of murine and human PGP.

9. Authors wrote that "In contrast, the CP2 analogs with the exception of 2E still retained inhibitory potency, with 2B being even slightly more potent than the parental compound." This data is not provided, please include in the SI.

The data are now provided in the table of the Supplementary Fig. 2B. For clarity, we have added another reference to this figure in the revised text.

10. It is needed to test CP1 against NT5C1A with the same concentration used for PGP inhibition. These assays have indeed been done in order to determine the IC_{50} , and we now show the concentration-response curves for CP1 and CP2 with NT5CA1, and for CP2 with NANP in the revised Supplementary Fig. 2D.

During preparation of the revised figure, we have noticed an issue with the employed curve fitting algorithm, which led to incorrect IC_{50} estimates for CP1- or CP2-treated NT5CA1. This has been corrected in the figure and the revised text (line 167): "CP1 and -2 inhibited the C1-capped soluble cytosolic 5'-nucleotidase 1A (NT5C1A), albeit with IC_{50} values ~57- or 14-fold above those for PGP".

11. Open and closed conformations of PGP are crucial, their superimposition should be (i.e. chains A-C, F and H) represented in a better way in Figure 2.

In the enlargements of the substrate selectivity loops and the flap domains in the revised Fig. 2C, we now show the superimposition of all monomers of the asymmetric unit (chains A-C, F and H) to better illustrate open and closed conformations. For greater clarity, we have chosen to show only chain A or chain E as representative open and closed conformers in the overview images that show the entire structure.

12. In line 221, the expression "excellent" should be removed from the sentence.
Removed.

13. In Figure 3B, the Asp34 should be represented and labeled at the figure.

We have now marked the position of Asn/Asp34 with a red asterisk in Fig. 3B (inhibitor map). Asn/Asp34 is shown in stick representation in Fig. 3C just below (detailed view of bound CP1 and adjacent residues of the active site); the view of the inhibitor bound to the active site in Fig. 3B and Fig. 3C is almost identical.

14. Chain-based RMSD values for CP1-bound PGP versus apo-PGP should be given.

Chain-based RMSD values (mean and standard deviation) are now reported along with the timeline graphs of the RMSD in Figure 4B.

15. Use RMSF plots to depict the fluctuations of the residues during molecular dynamics simulations.

RMSF plots are now provided in the Supplementary Figure 10.

16. Together with the Figure 4.A, which isn't clearly represented, preparation of a simulation movie could help readers to better understand the structural movement during the simulations.

Movies illustrating the movements along the first two principal components resulting from a PCA of the trajectories (cf. below, question 18) are provided as Supplementary Material (Supplementary Movie Files 1-4).

17. In Figure 4. A, representing the domains (cap and catalytic domain) by an arrow, will help with clarification.

We have clarified Figure 4 by indicating the cap domains with brackets and the core domains with a vertical bar. We have also included arrows showing the SSL and flap loop at the access to the active site.

18. Authors should also conduct PCA for the PGP simulations.

A Principal Component Analysis (PCA) has been conducted for the PGP simulations. The results are shown in Supplementary Figure 11.

19. Authors should also conduct binding free energy calculations (GBSA or PBSA) throughout the simulations.

and

20. In order to analyze the effect of individual binding energy contribution of the binding pocket residues on CP1 binding conduct per residue interaction analysis.

We did not carry out binding free energy calculations because we do not see any added value for the presented study. GBSA and PBSA only provide approximate estimates of (*prima facie* absolute, but ultimately relative) binding free energies and are particularly problematic for multiply charged compounds (as those used in this study). Also a partitioning into individual contributions is error prone and can be misleading. Instead, our manuscript reports precise experimental binding data and the results of *in vitro* mutational studies. We do not see any need to complement this with calculations faced with known limitations, nor would we like to expand our study towards a methodological work evaluating computational methods for free energy calculations.

21. To better understand the effect of the active residues to the binding of CP1, the *in vitro* mutational studies can be repeated by *in silico* simulations.

We neither see the need to carry out additional simulations to repeat the mutational studies *in silico*. The experimental data are clear and the mutation effects on CP1 binding can be readily explained from the binding mode of CP1 and the observed interactions in the crystal structure and the MD simulation. We have added the following statements on page 11 to clarify the structural interpretation of the mutation result: "These data are consistent with the binding mode of CP1: a replacement of Ser71 with Arg obstructs the binding site, a Glu at position 127 as well as an Asp at position 174 (both instead of a Pro) are electrostatically repulsive with respect to the distal carboxylate of CP1, and a Trp instead of Phe172 is sterically too demanding for CP1 binding. Figure 3D (right panel) shows that also the phosphatase activity of PGP^{S71R} was strongly decreased. This indicates a key role of Ser71 for active site accessibility."

22. Further discussion for the use of larger concentrations (100 μ M) in PGP-dependent cell proliferation and PGP-dependent droplet accumulation assay is needed.

To answer this question, we have performed mass spectrometric experiments (see Supplementary Methods). We have estimated intracellular CP1 concentrations to 0.13 μ M or 0.23 μ M upon incubation with 33 or 100 μ M CP1, respectively. These estimates are slightly below, but in any case in the range of PGP IC50 levels (CP1 IC50 is 0.39 μ M for human recombinant PGP *in vitro*; Fig. 1A).

As detailed in the Supplementary Notes and the Supplementary Discussion, it is likely that intracellular CP1 concentrations are somewhat underestimated, and that CP1 is subject to intracellular metabolism and/or degradation. Together, these results explain why relatively large CP1 concentrations are required in PGP-dependent cellular assays. In addition to CP1 metabolism/degradation, it is also possible that the cellular uptake of CP1 is limiting, and/or that CP1 is rapidly exported. We refer to the new data and discussion in the Supplementary Information of the revised manuscript (“Intracellular concentrations of CP1”).

23. Has CP1 previously been used on another target protein, if yes, it would need to be discussed and the activity should be linked to PGP inhibition.

We are not aware of any scientific publication describing CP1. A European Patent Application has been published in 2011 (Inventor: Grünenthal GmbH, EP 2332528A1; now ceased), in which substituted aromatic dicarboxylic acid amide compounds are claimed as inhibitors of the brain vesicular glutamic acid transporters vGLUT1 and vGLUT2 for the treatment of neuropathic pain. In this application, the compound we refer to as CP1 is reported to inhibit vGLUT1/2 up to 91% in vesicular glutamate uptake assays in vitro, with an IC₅₀ of 1.29 μM. The claim also covers an effectivity of CP1 in a rat neuropathic pain model. However, the patent application has been refused, and the underlying data remain unpublished. Because the reported activities cannot be evaluated, because nothing is currently known about the expression and potential role of PGP in neurons, and because VGLUT1/2 is specifically expressed in glutamatergic neurons, we do not discuss a possible activity of CP1 on VGLUT1/2.

24. Along with CP1; CP2 and -3 should also be used for MD simulations.

As mentioned above in reply to reviewer #2 (point 7), we have carried out docking calculations for CP2 and CP3 to compare their putative binding modes with CP1. Methods and results are provided as Supplementary Information.

25. Furthermore, together with CP1-3 simulations, a negative control simulation with weak PGP binders “among initially tested” could be performed.

We do not see any added value in this. Please note again that our work is not a computational molecular design study. There is no need for a negative control because we are not presenting a simulation method for compound selection or virtual screening.

26. MD simulations were carried out PGPD34N; C297S. Simulations should also be repeated for wild-type PGP.

As explained above, all MD simulation data in the manuscript have been updated to the simulations of PGP with wild-type active site. For this purpose, we replaced Asn34 with Asp and added the essential Mg²⁺ ion as seen in related phosphatases (details of this procedure are provided in the Methods section). The C297S mutation, introduced to abolish disulfide-mediated PGP tetramer formation, was maintained because it does not affect enzymatic activity and CP1 binding (as experimentally proven and mentioned in the manuscript) and is very unlikely to exert any effect on the dynamics of the PGP homodimer.

27. The system was protonated at pH 8.5 to imitate crystallization conditions, however authors need to test the effect of protonation states at physiological conditions.

Protonation at pH 7.5 (as determined by Protonate3D from MOE) did not lead to different protonation states.

28. The authors can expand their SAR study by searching diphenylether and diphenylmethylene dibenzoic acid derivatives in a more diverse library such as ZINC to obtain more hits or they may conduct this search at the FDA library for drug repurposing effort.

We thank the reviewer for this suggestion, which we will consider in future studies.

29. CP1 binding to the second binding site could be tested in silico.

In our opinion, binding of CP1 to the second binding site cannot really be “tested” in silico, unless perhaps with excessively long simulations or otherwise very demanding methods for estimating the

off-rate for dissociation of the ligand from its binding site(s). Given the available experimental data, which prove the relevance of the primary binding site and suggest a 1:1 stoichiometry of binding *in vitro* (unless the second binding event occurs with zero enthalpy), we cannot recognize the necessity to further investigate this crystallographic artifact with costly simulations.

30. In line 950, correct "Van-der-Waals" to "van der Waals".
Corrected.

31. Sequence identity between murine and human PGP should be added.
Indeed, all PGP residues found to engage in CP1 interactions are identical in murine and human PGP. The amino acid sequence alignment of murine and human PGP is now shown in the Supplementary Information (Supplementary Fig. 5).

Reviewer #4 (Remarks to the Author):

This manuscript reports the targeting of phosphoglycolate phosphatase (PGP) which is an important repair enzyme in glycolytic process. The study is novel in targeting a metabolite repair enzyme. The manuscript details an exhaustive and well-performed series of experiments using a broad tool-kit of techniques which ultimately support the idea that, since HAD phosphatases can show remarkable intrinsic substrate specificity and do not require an association with regulatory or targeting subunits to impart substrate selectivity, they represent a viable focus for future inhibitor development.

Overall I recommend publication of this work however, the following points should be considered in revision of the manuscript.

Table in Fig. 1 requires units.
We have added the unit in the revised table.

The reporting of thermodynamic parameters in the text and as a Table in Fig.1 is redundant without further analysis. For example, provision of the change in entropy data from ITC experiments for the two compounds does not mean anything without any reference to how these data describe the binding event, i.e. do they correlate with entropic features such as removal of water molecules or conformational change of the protein. In the absence of a meaningful structure/thermodynamic analysis the dissociation constant is the only value worth reporting.
The text has been modified (lines 131-133) and the binding enthalpies and entropies were removed from Fig. 1B.

The authors report that PGP exists in dynamic equilibrium between homodimers and tetramers. This could have significant implications in the lead compound binding studies. What has been done to mitigate against the possible impact of self-association of the enzymes. For example, is self-association enhanced by compound binding? Is there likely to be any allosteric effect between monomers of ligand binding?

PGP tetramers are only a minor species, and the dynamic equilibrium between PGP homodimers and tetramers is therefore unlikely to be a critical factor in our screening campaign/lead compound binding studies: (1) Using size exclusion chromatography-multi angle light scattering analysis, we have previously calculated PGP mass fractions to a tetramer/dimer distribution of 14/85% for PGP in the absence of a reducing agent (ref. 14). (2) PGP tetramer formation is redox-dependent and further decreases in the presence of a reducing agent to a tetramer/dimer distribution of 7/93% (ref. 14). (3) Reducing conditions are also required for maximal PGP activity (ref. 14). (4) Importantly, PGP tetramer formation does not affect the catalytic activity of the enzyme (ref. 14, and SI Fig. S3 of our present manuscript). Hence, it appears unlikely that compound effects on PGP self-association confound the results of the activity-based screening campaign.

We have taken the redox-dependent formation of PGP oligomers and the redox-dependent PGP activity into account by conducting the screening campaign in the presence of DTT as a reducing

agent. This is described in the Supplementary Methods section: “*High-throughput screening protocol*: (...) Measurements were conducted in the following assay buffer: Triethanolamine (30 mM), NaCl (30 mM), MgCl₂ (5 mM), **DTT (1 mM)**, Triton X-100 (0.01% v/v), pH 7.5.” Immediately after this sentence, we have now added the following text: “DTT was included as a reducing agent to suppress the redox-dependent formation of PGP oligomers and to ensure maximal PGP activity¹⁴.”

Using size exclusion chromatography, we have tested if CP1 binding affects PGP self-association. These experiments were done in the absence of a reducing agent to be able to observe both potential increases and potential decreases in PGP tetramer formation. As illustrated in the revised Supplementary Figure 3C, CP1 does not enhance self-association, but rather tends to decrease tetramer formation of PGP-WT. In contrast, no effect of CP1 on PGP-C297S self-association was observed. Like its closest homolog PDXP, PGP is an obligate, constitutive homodimer (ref. 22), and CP1 does not impact the homodimerization of PGP-WT or PGP-C297S.

We have revised the Results section as follows (lines 192-193): “We verified that PGP^{C297S} was enzymatically active and sensitive to compound 1 (PGP^{C297S} IC₅₀ ~0.25 ± 0.05 μM; **Supplementary Fig. 3A, B**), and that CP1 did not affect PGP^{C297S} self-association (**Supplementary Fig. 3C**; see below).

The crystal structural work is very well presented, however the authors report that “Towards the end of the refinement, residual density could be tentatively interpreted with a second CP1 molecule, which is binding to each monomer on the opposite side of PGP. The binding site is quite remote from the active site (closest distance of ~15 Å) and is in close proximity to the non-crystallographic twofold axis of symmetry. As the ITC data indicate binding of a single copy of CP1 to each monomer and the crystallographic evidence documents that the second CP1 molecule is weakly bound, no biological significance was attributed to the second binding site.” The assumption that, because the second binding event is not visible in the ITC data, it does not exist is incorrect. Some binding events have a zero (or negligible) change in enthalpy at a given temperature. Therefore, strictly speaking to be sure that the second binding event is weak or absent the ITC experiment should be done at a range of temperatures. I would not advocate this in this work, but some mention of the limitations of the experiment should be made.

The reviewer is certainly correct that the second binding event could be characterized by a small or zero change in enthalpy. We have edited the manuscript to account for this possibility (lines 266-270).

The cell-based assays focus on the human fibrosarcoma cell line, HT1080. The authors should provide justification for the choice of cell line.

We explain the choice of the HT1080 cell line in the revised Results section as follows (lines 353-356): “As a cellular model system, we used human fibrosarcoma HT1080 cells, a line whose malignant phenotype is driven by an activated N-ras oncogene that was established from a treatment-naïve patient²⁷. HT1080 cells are known to form highly aggressive tumors in murine xenograft models²⁸, suggesting potential translational relevance²⁹.”

In the revised Methods section, we have added the following text (line 682-683): “... HT1080 (a malignant human fibrosarcoma line established from a treatment-naïve patient), ...”

Clearly, the demonstration of activity of the compound across a range of cancer cells would provide convincing evidence for the efficacy of targeting of the phosphatase system. This should be mentioned as a limitation of the study.

We have added the following text to the last paragraph of the Discussion: “Future studies in *e.g.* a range of cancer cells and suitable disease models will need to evaluate whether the approach of blocking a metabolite repair protein such as PGP to control flux through the pentose phosphate pathway and glycolysis is effective with an adequate therapeutic index.”

After the initial screens and binding study the study focuses on CP1, despite the observation that CP2 and CP3 have similar PGP activity on PG and similar affinities for the target. The authors might want to comment on whether the other compounds could be investigated further.

We have included the following text in the Supplementary Discussion: “Although the present study focuses on CP1 as a prototypical PGP inhibitor, CP2 and its analogs such as 2B, as well as CP3 have similar PGP inhibitory activities and binding affinities (CP3). Although we have not yet conducted experiments beyond the ones reported in this study, CP2 and CP3 or their structural analogs are of interest and could be investigated further.”

REVIEWERS' COMMENTS

Reviewer #2 (Remarks to the Author):

In the revised version, the authors have added a number of new experiments and overall have done a very thorough job of addressing the points raised in the four reviews, which all had different sets of issues. There are no further concerns. The new CP1 phosphoglycolate phosphatase inhibitor could be a useful pharmacologically, particularly in cancer therapy.

Reviewer #4 (Remarks to the Author):

The Authors have addressed all of the issues that I raised on the original manuscript. Although there is still some concern about the impact of self-assembly on the data, this appears to be dealt with through adequate explanation in the revised version.

Reviewer #1 (Comments to the authors):

The authors have well added the NMR acquisition parameters as well as the (pre)-treatment parameters to the manuscript, which ensure that the experiments can be reproduced.

However, with regard to the quantification, none of the NMR pulse sequences applied offers the possibility to obtain absolute quantification without the use of calibration curves. The use of only one reference of a known concentration does not allow one to obtain absolute quantification, but only relative. However, in the manuscript, the precision that the concentrations reported are only relatives appears only once. Furthermore, the statement "Data were normalized by dividing the concentration of each metabolite by the cell number and by the absolute metabolite content of the respective sample; this value was multiplied with the group average of the absolute metabolite content." brings even more confusion in the fact that the quantitation reported are only relative.

We have now added explanatory sentences and an additional reference (ref. 67) to avoid any confusion (changes and additions to the revised text are marked in red; please see below).

Quantitative NMR provides absolute quantification in the sense that we do know the concentration (*i.e.*, gram/gram) of a chemical compound dissolved in a solution. Usually, qNMR does not require calibration curves based on different concentrations of a calibrating compound, because only one specific type of nucleus (*e.g.*, H, C, P) is measured in an experiment and therefore, response factors are equal in all molecules and in a fixed ratio between different compounds defined by the number of protons within the molecule. We would gain from a calibration curve in terms of precision; however, in the context of a biological sample this would be a minor gain relative to other uncertainties (*e.g.*, the number of cells).

With regard to the data normalization: We are still working with absolute concentrations. In the first step, the concentration per cell is calculated, then multiplied and divided by a concentration. At the end of the procedure, we still maintain a concentration, however, in a statistical sense this is a weighted concentration, which could be called a "relative" concentration (relative to the group average). However, we believe that changing the naming to "relative concentration" will be confusing, as it is relative in a statistical sense and not in the sense of an analytical measure.

The revised Methods part now reads:

A 9 mmol/L aqueous sucrose solution was used as a calibration compound to calibrate an ERETIC2 reference for absolute concentration determination^{66,67}. Based on the sucrose reference, an electronic signal was placed in each ¹H spectrum to calibrate the deconvolution software Chenomx NMR suite. A standard protocol for quantitative NMR was applied: The signal corresponding to the calibration compound was integrated and normalized according to the number of protons of this compound. The unknown concentration of compound *x* in the presence of the calibration compound is calculated according to:

$$C_x = \frac{I_x}{I_{cal}} \frac{N_{cal}}{N_x} C_{cal}$$

With I, N, and C being the integral area, number of nuclei, or concentration of the calibration (*cal*) or the targeted compound (*x*), respectively.